# DEEP REFLECTION HINTING: LEVERAGING OFFLINE KNOWLEDGE FOR IMPROVING WEB AGENTS ADAPTATION

## ABSTRACT

Large language model (LLM) agents perform well in sequential decision-making tasks, but improving them on unfamiliar domains often requires costly online interactions or fine-tuning on large expert datasets. These strategies are impractical for closed-source models and expensive for open-source ones, with risks of catastrophic forgetting. Offline trajectories offer reusable knowledge, yet demonstration-based methods struggle because raw traces are long, noisy, and tied to specific tasks. We present *Deep Reflection Hinter (*DR. HINTER*)*, an agentic system that distills offline traces into compact, context-aware hints. A zooming mechanism highlights decisive steps in long trajectories, capturing both strategies and pitfalls. Unlike prior methods, DR. HINTER leverages both successful and failed trajectories, extracting guidance even when only failure data is available, while supporting parallelized hint generation and benchmark-independent prompting. At inference, a retriever selects relevant hints for the current state, providing targeted guidance with transparency and traceability. Experiments on MiniWoB++, WorkArena-L1, and WebArena-Lite show that DR. HINTER consistently outperforms strong baselines, including human- and document-based hints.

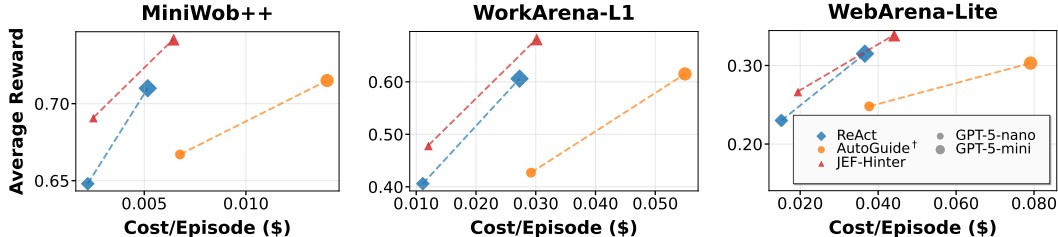

Figure 1: Average episodic reward versus test-time evaluation cost of DR. HINTER on MiniWoB++, WorkArena-L1, and WebArena-Lite, using **GPT-5-mini** as the Hinter model. Colors and markers denote different methods, while marker size reflects the base LLM model size.. DR. HINTER achieves substantial gains over baselines, incurring only slightly higher cost than the original ReAct (Yao et al., 2023b) agent while being far more efficient than Autoguide[†] (Fu et al., 2024).

## 1 INTRODUCTION

Large language model (LLM) agents have shown impressive abilities in sequential decision-making tasks such as web navigation and interactive environments. Yet their performance often deteriorates in unfamiliar domains due to incomplete domain knowledge and reasoning gaps. Unlike static tasks, sequential settings amplify small mistakes, where an early error can derail the entire trajectory. Offline resources offer an attractive opportunity. Trajectories from prior agents (both successful and failed), human demonstrations, and organizational documents all encode reusable decision patterns. Leveraging this knowledge is particularly important for closed-source models, which cannot be fine-tuned, and for large open-source models, where fine-tuning is costly and often risks catastrophic forgetting. Methods that can distill reusable knowledge from offline data provide a scalable way to improve state-of-the-art models without retraining or waiting for new releases.

Supervised fine-tuning on offline trajectories can appear to work, but off-policy bias means the learned policy cannot reliably execute even the training tasks end-to-end on its own, and it generalizes poorly to new tasks (Ouyang et al., 2022; Yao et al., 2022). Reinforcement learning can be effective for web agents(Vattikonda et al., 2025), but its reliance on extensive online interactions is impractical at scale, and it cannot be applied to closed-source models. Retrieval-augmented generation (RAG) methods, such as in-context demonstrations (Lewis et al., 2020), provide task-specific examples at inference, but raw trajectories are long, noisy, and tightly bound to their source tasks, limiting transfer. Recent work, such as AutoGuide (Fu et al., 2024), addresses part of this gap by distilling guidelines from offline trajectories, but it is limited to contrastive trace pairs and uses benchmark-specific prompting. These challenges motivate a more general and scalable framework for extracting and reusing offline knowledge.

We introduce DR. HINTER, an agentic system that distills offline traces into explicit, context-aware hints. Instead of replaying full trajectories (Shinn et al., 2023; Fu et al., 2024), DR. HINTER employs a *zooming module* to focus on critical decision points and a *reflection step* to convert them into concise natural-language hints capturing both effective strategies and common pitfalls. Hints can be generated from single traces, pairwise contrasts, or multi-trace aggregation, ensuring coverage even when no successful run exists. Each hint is paired with a *semantic key* for retrieval, enabling either fine-grained step-level guidance or efficient goal-conditioned retrieval at inference preventing overload from irrelevant information (Zhao et al., 2024) and complementing intra-task reflection mechanisms (Shinn et al., 2023). This offline-to-online pipeline produces a lightweight database of actionable hints that improves agent robustness and long-horizon generalization without requiring model fine-tuning. Since DR. HINTER represents guidance as explicit hints linked to their source traces or documents, it provides greater transparency and traceability than both supervised fine-tuning and in-context RAG, allowing systematic analysis of how offline data influences agent behavior.

**Contributions:**

- We introduce *Deep Reflection Hinter (*DR. HINTER*)*, an agentic system that distills offline trajectories into explicit, context-aware hints. DR. HINTER features parallelized hint generation, intelligent zooming on critical steps, and flexible trace selection (single, pairwise, or multi-trace), leveraging both successful and failed runs.

- We evaluate DR. HINTER across MiniWoB++, WorkArena-L1, and WebArena-Lite, where it consistently outperforms strong baselines. We further compare against documentation retrieval and human-authored hints, showing that automatically generated hints provide more scalable and broadly effective guidance.

- We provide qualitative analyses that illustrate how DR. HINTER addresses common agent failure modes by steering actions toward the correct context and preventing repeated errors, thereby improving robustness and transparency.

## 2 RELATED WORK

LLMs have shown strong reasoning capabilities (Wei et al., 2022), resulting in LLM-based agents applied on a variety of real-world interactive tasks, including web navigation (Nakano et al., 2021; Wei et al., 2025; Zhang et al., 2025). However, performance on multiple web-focused benchmarks (Yao et al., 2022; Deng et al., 2023; Zhou et al., 2024b; Koh et al., 2024; Drouin et al., 2024; Boisvert et al., 2024) indicates that, as-is, LLMs still struggle with complex tasks requiring planning over long horizons. This gap has motivated several directions of work on improving LLM-based agents.

**Prompting and reflection.** A large body of work explores prompting strategies to elicit stronger reasoning and planning from LLMs. ReAct (Yao et al., 2023b) interleaves reasoning steps with environment actions to structure trajectories. Building on this, Reflexion (Shinn et al., 2023) introduces self-reflection over past trials to refine behavior, while ExpeL (Zhao et al., 2024) mines offline Reflexion trajectories to extract reusable skills. Other approaches focus on explicit planning: Ada-Planner (Sun et al., 2023) iteratively adapts a plan to specific task instances, and AutoPlan (Ouyang & Li, 2023) instead optimizes for generalizable plans across instances. Methods such as Inner Monologue (Huang et al., 2023) and Self-Refine (Madaan et al., 2023) further extend reflection by

continuously revising intermediate reasoning. A concurrent work, Atomic Fact Augmentation with Lookahead Search (Holt et al., 2025), enhances in-context planning via fact extraction and local search but targets short-observation domains like ALFWorld, making it complementary to our focus on large web-scale observations.

**Search-based planning.** Beyond prompting, several works integrate symbolic search with LLM reasoning to better handle long-horizon tasks. Tree-of-Thoughts (Yao et al., 2023a), Language Agent Tree Search (Zhou et al., 2024a), and their variants (Putta et al., 2024; Koh et al., 2025) explore branching reasoning paths and dynamically selecting among them, improving robustness on tasks where single-line chain-of-thought often fails. While effective, these approaches typically require large test-time compute budgets and do not leverage offline knowledge.

**Offline data and hinting.** Orthogonal to online prompting and search, another line of work focuses on extracting reusable guidance from offline data. RAG approaches (Lewis et al., 2020) have been adapted for agents by retrieving demonstrations or examples (Yao et al., 2023b), but raw trajectories are long, noisy, and task-specific, limiting their transferability. AutoGuide (Fu et al., 2024) addresses part of this challenge by distilling guidelines from contrastive trajectory pairs, showing that abstracted guidance can outperform raw demonstrations. More recently, Agent Workflow Memory (AWM) (Wang et al., 2024) induces reusable workflows from successful trajectories, enabling agents to accumulate and reuse subroutines across tasks. AutoManual (Chen et al., 2024) similarly aims to build reusable manuals but requires online interaction and repeated rule revision, whereas our approach works fully offline and supports cross-task, cross-agent hint synthesis.

However, both AutoGuide and AWM remain constrained in scope: the former requires contrastive pairs, while the latter depends on successful traces alone. By contrast, our approach extracts hints from both successes and failures, making them more general than workflows or contrastive guidelines. Hints capture not only reusable strategies but also common pitfalls, providing broader and more flexible guidance. We further enable parallelized extraction for scalability and integrate heterogeneous offline sources such as domain documents and human-written instructions into a unified framework.

## 3 DEEP REFLECTION HINTING

Large language model (LLM) agents often struggle to generalize across tasks when relying solely on their base policy $\pi$. Direct fine-tuning can be costly, unstable, or even impossible for closed-source models. To address this, we propose to improve $\pi$ by supplying it with targeted, reusable guidance extracted from offline experience. At the center of our approach is the *Hinter $\mathcal{H}$*, itself an LLM, a model that transforms trajectories and documents into explicit natural-language hints. Since hint generation is performed offline, $\mathcal{H}$ can be significantly larger and more capable than the base agent, yet the resulting hints remain lightweight at inference. We instantiate this method as DR. HINTER, which systematically augments the LLM base policy with retrieved hints to enhance decision making without any fine-tuning.

### 3.1 DATA COLLECTION

Unlike prior work such as AutoGuide (Fu et al., 2024), which extracts guidance only from contrastive trajectory pairs, DR. HINTER can operate over a broader range of offline signals. Given a dataset of trajectories $\{\tau_1, \ldots, \tau_N\}$, it flexibly selects evidence for hint generation. The trajectories may come from the base policy $\pi$, which yields hints tailored to its strengths and weaknesses, but they can also originate from other agents or human demonstrations. We support three complementary modes:

1. *Single-trace analysis.* Generate hints from a single trajectory $\tau$, highlighting effective decisions in successful segments and exposing pitfalls in failed ones.

2. *Pairwise analysis.* Contrast two trajectories $(\tau^+, \tau^-)$ where the total reward assigned to $\tau^+$ is greater than the reward assigned to $\tau^-$, and identify the key divergences that explain the performance gap. If no such pair is available, we also allow equal-reward or (fail, fail) and (success, success) pairs.

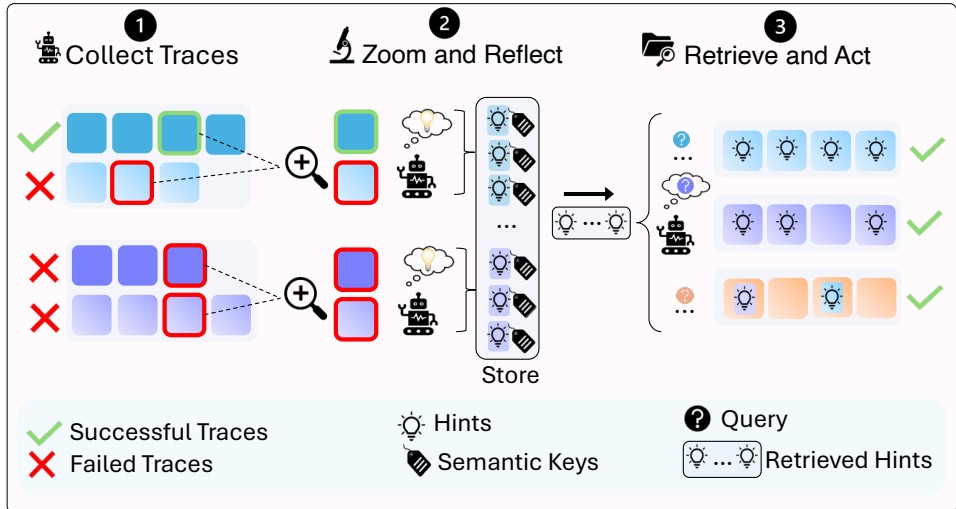

Figure 2: Overview of the DR. HINTER. **(1) Collect Traces:** DR. HINTER operates over heterogeneous offline trajectories, including both successful (green) and failed (red) runs, allowing the system to capture not only effective behaviors but also common pitfalls. **(2) Zoom and Reflect:** A zooming module selects critical steps within each trace, and the hinter reflects on these segments to distill them into concise, reusable natural language hints. Each hint is paired with a semantic key summarizing its context and stored for retrieval. **(3) Retrieve and Act:** At inference time, the agent generates a query (goal- or context-conditioned) which is matched against the database of semantic keys. The most relevant hints are retrieved and injected into the agent's context, guiding its actions. This process unifies knowledge distillation, reflection, and retrieval, supporting both in-task reliability and out-of-task generalization.

    3. *Multi-trace analysis.* Combine a set of trajectories $\{\tau^i\}_{i \in S}$ to surface patterns that are robust across instances and transferable across tasks.

## 3.2 HINT GENERATION: ZOOM & REFLECT

A trajectory provides four types of signals: observations $x$ such as screenshots, HTML (or AxTree); reasoning tokens $z$ that record intermediate thoughts; actions $a$ that alter the environment; and rewards $r$ that measure progress. The initial observation $x_0$ also contains the goal $g$. We combine these signals to form the prompt $(P)$ given to the Hinter. The simplest option is the full prompt $P_\tau^{\text{full}} = \{x, z, a, r\}_{1:T}$, which passes the entire trajectory as context. Long-horizon tasks quickly make this representation unwieldy. To address this, we introduce a Zooming LLM module that selects critical steps $t^*$ and extracts a compact prompt:

$$P_\tau^{\text{zoom}} = \{z, a, r\}_{1:T} \; \cup \; \{x\}_{t^*:t^*+\Delta}.$$

This keeps the full sequence of reasoning, actions, and rewards, while restricting observations to the decisive windows. The parameter $\Delta$ specifies the length of the observation window appended after $t^*$, determining how much context is retained. Critical steps correspond to points where the agent makes an important choice, repeats a common mistake, executes a successful strategy, interacts with a key element, handles a timing dependency, or reaches a definitive outcome. For instance, in a web form task, repeatedly clicking the wrong navigation bar is flagged as a critical step, while in a multi-select list, the decisive step is holding Ctrl/Cmd to select multiple items. Appendix C.1 details the step-selection procedure.

Next, to support retrieval, we generate a *semantic key* summarizing the trajectory prefix. Given $\tau_{:t}$, the summarizer $\mathcal{S}$ outputs a short natural-language context $c_t = \mathcal{S}(\tau_{:t})$. This key anchors hint generation during training and enables efficient lookup at inference.

Finally, given a context $c_t$ and a prompt $P_\tau$, the Hinter produces a hint

$$h = H(c_t, P_\tau),$$

which captures either a beneficial action or a common error to avoid. We collect all hints in a database $\mathcal{D}_{\mathcal{H}} = \{(c_t, h)\}$, linking each hint to the semantic key from which it was derived (see Appendix F for pseudocode).

## 3.3 RETRIEVE & ACT

We explore two complementary strategies for retrieving and applying hints during inference.

**Contextual retrieval with step-level hints.** At each time step $t$, the summarizer produces a context $c_t = \mathcal{S}(\tau_{:t})$. The retrieval LLM module $\rho$ then selects the top $k$ hints most relevant to that context, $\{h_t^1, \ldots, h_t^k\} = \rho(c_t, \mathcal{D}_{\mathcal{H}})$, and the policy conditions its next action on both the trajectory prefix and the retrieved hints, $a_t \sim \pi(x_{0:t}, \{h_t^1, \ldots, h_t^k\})$. This approach provides fine-grained, context-specific guidance, but it is computationally costly since it requires one model call to establish the context and retrieve hints and another to generate the action.

**Goal-conditioned retrieval with episode-level hints.** A more efficient strategy retrieves hints once at the start of an episode, using the goal $g$ as the retrieval context: $\{h^1, \ldots, h^k\} = \rho(g, \mathcal{D}_{\mathcal{H}})$. The policy then acts while simultaneously selecting a relevant hint from this fixed set, $(a_t, h_t) \sim \pi(x_{0:t}, \{h^1, \ldots, h^k\})$. This method avoids repeated retrieval calls and reduces inference cost, while still maintaining sufficient contextual relevance.

**Source tasks for retrieval** The choice of source tasks also determines how well hints generalize. In-task retrieval draws hints from the same task but with different goals[1], which strengthens reliability within a domain. Cross-task retrieval excludes the source task altogether and forces the agent to transfer knowledge from other tasks. Hybrid retrieval mixes both approaches with adjustable weighting, striking a balance between reliability and transfer. Because hints capture abstract decision patterns rather than raw demonstrations, they remain effective across goals and tasks under both settings.

The zooming module, summarizer $\mathcal{S}$, hinter $\mathcal{H}$, retriever $\rho$, and base policy $\pi$ are all LLM-based components. In contrast, the hint database $\mathcal{D}_{\mathcal{H}}$, its indexing and storage, and the embedding-based matching used for retrieval are lightweight non-LLM operations responsible for orchestration and lookup.

## 4 EXPERIMENTAL SETUP

**Benchmarks** We evaluate on three widely used benchmarks that span increasing levels of complexity: MiniWoB++ (Liu et al., 2018), a suite of synthetic single-page UI tasks; WorkArena-L1 (Drouin et al., 2024), a benchmark of enterprise knowledge-work tasks involving multi-step form filling and navigation; and WebArena (Zhou et al., 2024b), a realistic environment of multi-domain web tasks requiring long-horizon reasoning. Together, these benchmarks test both short-horizon precision and long-horizon generalization.

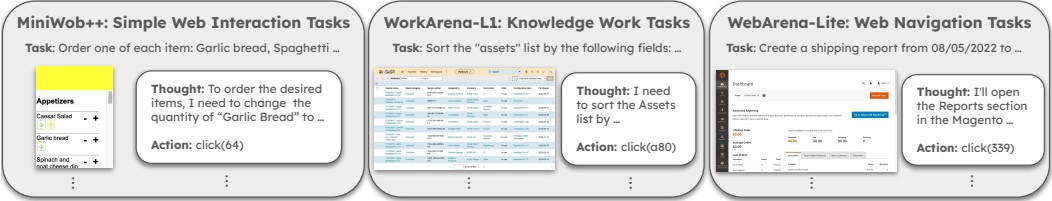

Figure 3: Web browsing benchmarks considered in our work: MiniWob++ (Liu et al., 2018), WorkArena-L1 (Drouin et al., 2024), and WebArena-Lite (Zhou et al., 2024b; Liu et al., 2025).

---

[1]Benchmarks like MiniWoB++ and WorkArena support multiple seeds per task. We refer to a specific instance of a task as a goal.

**Observation and action spaces**  To improve speed and efficiency, we work with the accessibility tree (AXTree). This reduces the size of the input by about 10x compared to the HTML DOM trees. Exceptionally, on MiniWoB++, we work directly with the DOM since it is small enough and contains more of the relevant information. The action space across all environments consists of high-level UI primitives such as `click(node)`, `fill(node, text)`, `select(node, option)`, `scroll(node)`, and `hover(node)` as provided by BrowserGym. This abstraction enables consistent evaluation across benchmarks with differing interfaces.

**Baselines**  All methods build on the ReAct agent framework (Yao et al., 2023b), which combines chain-of-thought reasoning with environment interaction. We compare against: (i) **ReAct** without offline hinting, (ii) Our implementation of **AutoGuide** (Fu et al., 2024), which augments ReAct with offline guideline extraction from contrastive trajectory pairs. We call this agent AutoGuide[†]. In addition, we evaluate two variants of our agent: **DR. HINTER** (w/o zoom), our basic implementation that takes the full trajectory as input and distills offline trajectories into natural-language hints (for WorkArena-L1 and WebArena-Lite, we drop AxTrees to fit within the hinter model's context), and **DR. HINTER**, which further includes zooming on critical steps.

**Offline datasets**  We construct offline datasets using the AGENTLAB framework (Drouin et al., 2024; Chezelles et al., 2025). For MiniWoB++, we collect trajectories by running a ReAct agent on 5 held-out goals per task, and for WorkArena-L1, we collect trajectories on 10 held-out goals per task. For WebArena, we use WebArena-Lite Liu et al. (2025) for parallel trace collection. In all benchmarks, we retain both successful and failed trajectories so that hint extraction can cover both positive decision points and common pitfalls. In contrast, AutoGuide (Fu et al., 2024) requires pairs of successful and failed traces and therefore only produces hints when both are available. To study the impact of dataset quality, we additionally construct augmented datasets by including traces from GPT-5, ensuring at least one successful trace per task. If no successful trace exists, even after augmentation, AutoGuide produces no hint for that task, whereas DR. HINTER can still generate useful hints from failed trajectories alone.

**Evaluation protocol**  We evaluate generalization under two complementary settings: *In-task generalization:* The agent retrieves hints only from the same source task, but from different goals than those used in evaluation. This setting measures how well hints transfer within a task across different environment initializations. *Out-of-task generalization:* To assess a more challenging scenario, we exclude the source task entirely from the hint database. At inference time, the agent must instead rely on hints retrieved from other tasks, using the LLM retriever or embedding vector matching to select the most relevant ones. This setup tests whether hints distilled from one set of tasks can transfer effectively to unseen tasks with different structures.

The primary evaluation metric is average task success rate, reported separately for in-task and out-of-task settings. We also provide qualitative analysis of retrieved hints to illustrate their interpretability and usefulness.

## 5 EMPIRICAL STUDY

We present results through research questions examining the effectiveness, generalization, and design decisions of DR. HINTER.

### 5.1 DOES DR. HINTER IMPROVE OVERALL PERFORMANCE COMPARED TO BASELINES?

To address this question, we compare ReAct, AutoGuide, and DR. HINTER across three benchmarks: MiniWoB++, WorkArena-L1, and WebArena-Lite. As shown in fig. 4, three key findings emerge:

**Generally, hints provide effective guidance to agents.**  Both AutoGuide and DR. HINTER consistently outperform vanilla ReAct across all benchmarks and base models. This confirms that offline hints provide meaningful guidance, steering the agent away from common pitfalls and toward

---

[1][†]Since no public implementation of AutoGuide was available, we re-implemented it within our ReAct framework for consistency and comparability.

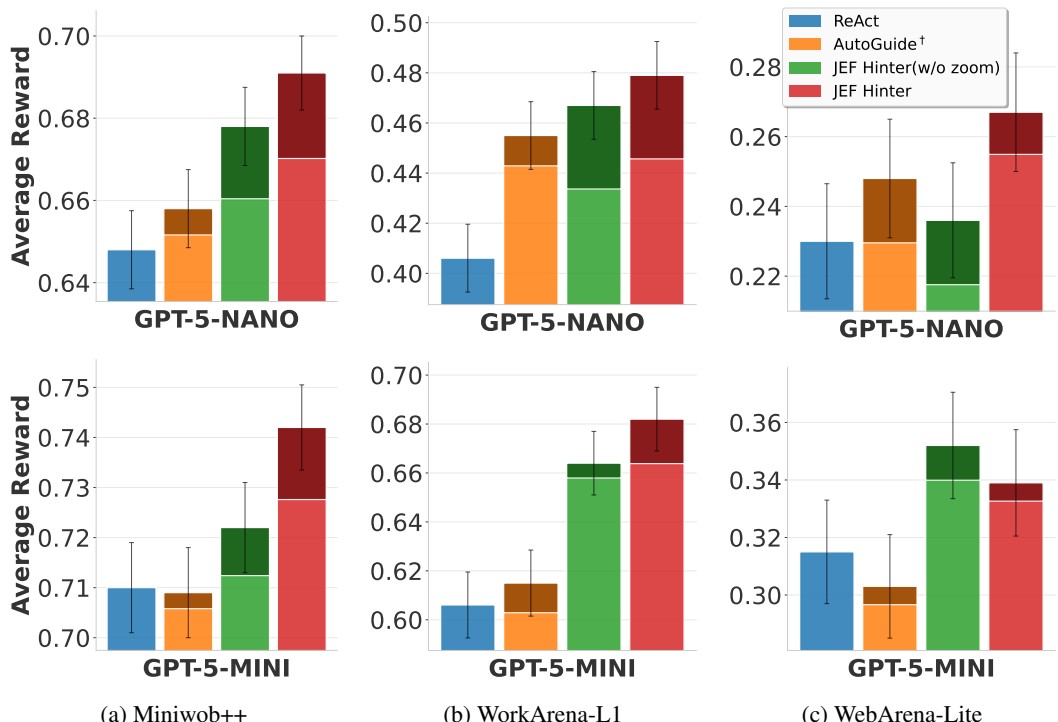

(a) Miniwob++      (b) WorkArena-L1      (c) WebArena-Lite

Figure 4: Average reward comparison across MiniWoB++, WorkArena-L1, and WebArena-Lite using two base models with **GPT-5-mini** as the Hinter model. DR. HINTER and DR. HINTER(w/o zoom) consistently outperform all baselines across most tasks, highlighting the effectiveness of our approach. Shaded regions denote tasks where the base ReAct agent failed entirely, highlighting DR. HINTER's ability to extract useful hints even from failure-only trajectories.

more successful strategies. Moreover, since GPT-5-mini is used as the hinter model, the gains observed when the base model itself is GPT-5-mini highlight that DR. HINTER enables effective *self-improvement*, demonstrating that a model can refine its own decision-making by reflecting on past traces.

**Even failed trajectories can provide constructive hints.** While AutoGuide improves performance over ReAct, its gains are larger for weaker base models and often limited to relatively simple hints due to its reliance on contrastive pairs. In contrast, DR. HINTER outperforms AutoGuide by generating hints from *all* available trajectories—successful or failed—rather than only paired traces. This flexibility allows

Table 1: Ablation of full-trace vs. zoomed multi-trace hinting.

| Method | MiniWoB++ | WorkArena-L1 |
|---|---|---|
| ReAct | 0.715 | 0.661 |
| DR. HINTER (FT) | 0.718 | 0.715 |
| DR. HINTER | 0.739 | 0.770 |

DR. HINTER to extract actionable guidance even from failure-only data, leading to higher task performance. To emphasize this, we report performance on tasks where the baseline ReAct agent failed entirely, shown as darker bars in fig. 4.

**Entire trajectories are not always necessary for high quality hints.** DR. HINTER improves over its non-zooming variant by selectively surfacing the most critical steps from each trajectory. To better isolate the role of this mechanism, we additionally compare DR. HINTER to a variant that provides the hinter with the *entire* execution trace—every observation, action, and think token—without any step selection. Using GPT-5 for both the base agent and the hinter, supplying

a full single trace yields only marginal gains over ReAct. In contrast, zooming over two trajectories produces substantially larger improvements on both MiniWoB++ and WorkArena-L1 (Table 1). These findings indicate that the hinter benefits not from receiving more context, but from receiving more *informative* context: the ability to compare trajectories and focus on high-salience AXTree snapshots is essential for generating actionable hints. Since zooming is performed entirely offline, these gains come at no additional inference-time cost.

**Stronger base and hinter models.** We further evaluate DR. HINTER with stronger base agents and stronger hinter models to assess whether hint-based adaptation remains effective with state-of-the-art models. Using GPT-5 as both the underlying ReAct agent and the hinter, DR. HINTER improves performance from 0.715 to 0.739 on MiniWoB++ and from 0.661 to 0.770 on WorkArena-L1. We also experiment with a vision-based CUA agent paired with Claude-4.5-Sonnet serving as both the base model and the hinter, observing gains from 20.6% to 40.0% on WorkArena-L1 and from 27.3% to 29.1% on WebArena-Lite. These results demonstrate that DR. HINTER continues to provide meaningful benefits even when applied to powerful modern models.

## 5.2 How effective is Dr. Hinter compared to documentation and human hints?

**Alternative sources of guidance.** To assess the value of trajectory-based hints, we compare DR. HINTER against two alternative sources: platform documentation and human-authored instructions. Unlike DR. HINTER, these hints are not distilled from trajectories but taken directly from raw resources—documentation webpages or short annotator notes—and retrieved at inference time. This comparison tests whether explicit external guidance can match or exceed the utility of trajectory-derived hints.

**Baseline configurations.** For documentation, we collected platform-specific materials: ServiceNow for WorkArena-L1, and GitLab and Shopping sites for WebArena. Pages were retrieved with BM25 using the task goal as the query, and the top-ranked passages were provided directly to the agent as hints (see appendix A for details). Human hints were prepared only for WorkArena-L1: we curated concise notes for 16 particularly challenging goals, covering all task types while focusing on cases where automated hinting failed. In both baselines, the retrieved content was used as a direct substitute for trajectory-based hints, not in combination. Results of these comparisons are reported in table 2, with details of the human hint collection in appendix B.

Table 2: Comparison of DR. HINTER against alternative hinting strategies. Results are reported as average reward with standard error of 0.01 on WorkArena-L1 and 0.03 on WebArena-Lite.

| Method | WorkArena-L1 | WebArena-Lite |
|---|---|---|
| **GPT-5-NANO** | | |
| ReAct | 0.41 | 0.23 |
| Human hints | 0.43 | – |
| Documentation | 0.44 | 0.20 |
| DR. HINTER | **0.48** | **0.27** |
| **GPT-5-MINI** | | |
| ReAct | 0.61 | 0.32 |
| Human hints | 0.66 | – |
| Documentation | 0.64 | 0.33 |
| DR. HINTER | **0.68** | **0.34** |

**Effectiveness of external resources.** External resources can substitute for trajectory-based hints, but with notable trade-offs. Documentation retrieval scales easily and provides modest gains, though its utility depends heavily on manual quality and often yields only partially relevant context. Human hints (limited to 16 curated goals), while effective are expensive to obtain and hard to scale. Overall, both baselines help bridge knowledge gaps, but DR. HINTER is more practical: it automatically produces reusable hints from offline traces without relying on manuals or human annotation.

## 5.3 Can Dr. Hinter generalize out-of-task?

To assess out-of-task generalization, we remove the source task used to generate hints from the retrieval pool. The retriever must then select the most relevant hints by matching the current task goal against the remaining database entries. As shown in fig. 5, DR. HINTER sustains competitive performance under this setting, indicating that trajectory-derived hints can transfer beyond the tasks

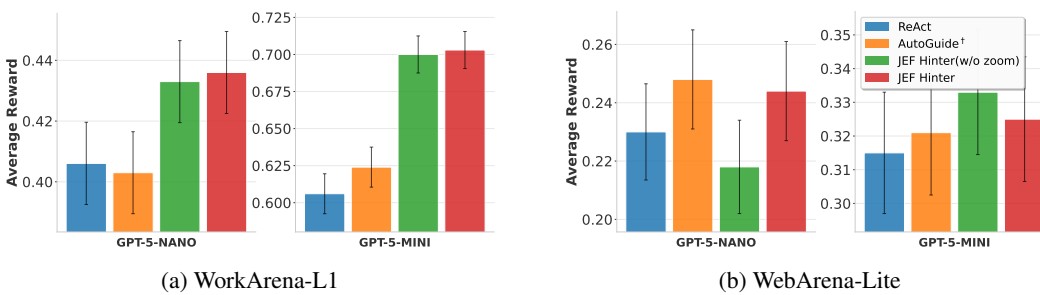

(a) WorkArena-L1                  (b) WebArena-Lite

Figure 5: Out-of-task generalization performance on WorkArena-L1 and WebArena-Lite using two base models with **GPT-5-mini** as the base for the hinter model.

they were trained on. On WorkArena-L1, we still observe clear gains over both ReAct and AutoGuide, while on WebArena-Lite, all methods perform within the margin of noise, suggesting that this benchmark remains especially challenging for cross-task transfer.

Regarding broader cross-benchmark generalization (e.g., using MiniWoB++ hints for WorkArena or WebArena tasks), we note that hint extraction distills environment-specific UI structures and interaction patterns. As a result, transferring hints across benchmarks with different element vocabularies, accessibility-tree formats, and task types can introduce mismatches rather than clear benefits. This is a common limitation for UI-grounded memory and retrieval approaches. We added a paragraph to section 5.3 clarifying the intended scope of DR-HINTER: efficient offline adaptation within web domains, rather than universal cross-environment transfer.

## 5.4 ANALYSIS & DISCUSSION

**How does the size of the hinter model affect performance?**

Figure 4 showed that GPT-5-mini can already serve as a capable hinter for both GPT-5-nano and itself. To isolate the effect of capacity, we ablate the hinter model from GPT-5-mini to GPT-5. As shown in Figure 6, the larger hinter generally produces higher-quality hints, translating into stronger downstream performance. Gains are most pronounced on complex, long-horizon tasks such as WorkArena-L1($+5\%$), where fine-grained context understanding and precise hint phrasing matter most. On simpler tasks like MiniWoB++ ($+2\%$), the advantage narrows, suggesting that larger hinters are particularly useful when reasoning demands are high. Thus, scaling the hinter model improves performance but introduces a clear trade-off between quality and computational cost.

**Qualitative analysis.** Case studies illustrate how DR. HINTER's hints intervene precisely at the decision points that previously caused failures, directly correcting the agent's reasoning and enabling successful task completion.

**MiniWoB++.** In the `click-scroll-list` task, the agent is instructed to "Select Bermuda, Saint Lucia from the scroll list and click Submit." Without hints, the agent frequently fails because it clicks the items sequentially without holding the control key, which causes earlier selections to be deselected. A relevant retrieved hint states: *"In a multi-select scroll list, hold Ctrl (Cmd on Mac) and click each required item so all stay highlighted, then click the Submit button."* This explicit correction allows the agent to overcome the failure mode of not performing multi-selection. Refer to appendix E.2 for the full reasoning and output of the DR. HINTER agent.

**WorkArena-L1.** In the `filter-navigation` tasks, the ReAct baseline often failed by relying on the wrong search context (e.g., the global bar or the *Workspaces* filter) or by clicking too early before the application menu expanded, causing repeated loops without progress. DR. HINTER corrected these errors by providing an explicit hint to use the Application Navigator's *All* menu, enter the application name in the correct filter box, and wait for the menu to expand before clicking

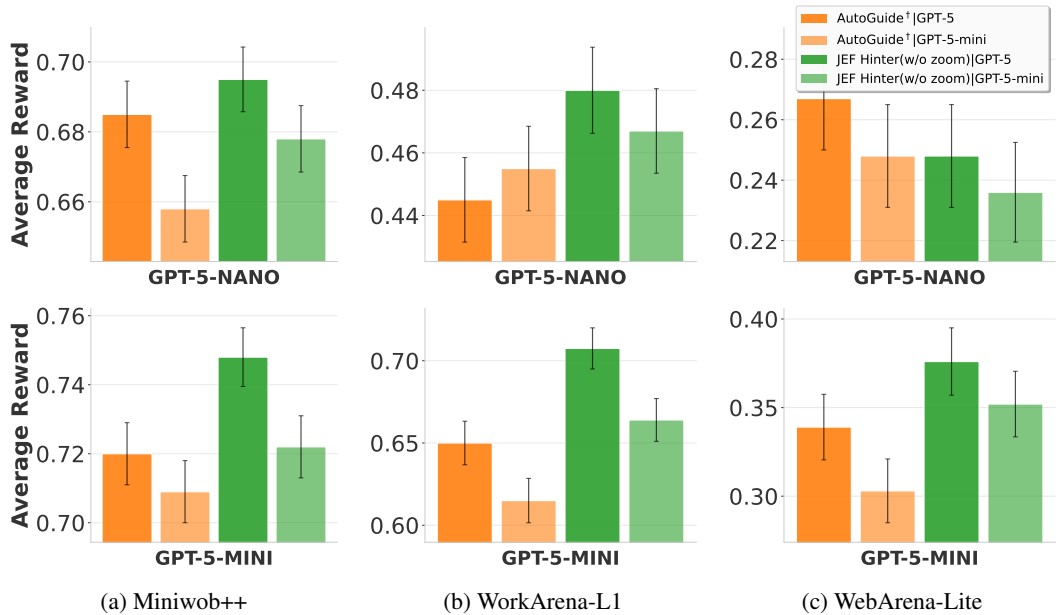

Figure 6: Comparison of hinter models (GPT-5-mini vs. GPT-5) on MiniWoB++, WorkArena-L1, and WebArena-Lite. Larger hinters generally provide higher-quality hints, with the biggest gains on complex, long-horizon tasks.

the target module. With this guidance, the agent consistently reached the intended *Active* module, avoiding wasted actions and navigation errors. Refer to appendix E.2 for the full reasoning and output of the DR. HINTER agent.

**WebArena-Lite.** In the Shopping Admin tasks, the agent must identify the customer with the most cancellations over the entire history. The *generic* ReAct agent often failed by relying on surface inspection of the first page of results and answering from what was visible without using the grid's controls. It did not open Filters, left a default date restriction in place (thus undercounting "history"), sometimes relied on keyword search or *Advanced Reporting*, and neither sorted nor paginated to aggregate counts, leading to incorrect totals. By contrast, DR. HINTER followed the detailed sequence provided by the following hint: *Go to Sales > Orders, open 'Filters', set 'Status' to 'Canceled', click 'Apply Filters', clear the 'Search by keyword' box, then sort the 'Bill-to Name' column to group names and scan/paginate for the largest group; to verify counts, use the 'Bill-to Name' filter and read 'records found', removing that chip before testing another; avoid 'Advanced Reporting'.* By closely following this sequence of steps, the agent is able to complete the task successfully. Figure 11 in appendix E.2 shows how the hinted agent leverages provided hints to properly select the right action to take in order to solve a task.

Case Study for Zooming Mechanism (workarena.servicenow.sort-hardware-list task). This task shows the strongest impact of zooming: DR-Hinter reaches only 10% success without zooming, but rises to 70% when zooming is enabled—even though the base agent is identical. The gain comes entirely from higher-quality hints generated when the hinter is given zoomed AXTree snapshots. Zooming surfaced four decisive steps drawn from both a successful trajectory (Steps 4 and 8) and a failed one (Steps 14 and 16). Step 4 captures the moment the agent selects the primary sort field and exposes the full dropdown; Step 8 contains a clean post-action AXTree after applying the sort; and Steps 14 and 16, although from a failing run, reveal the same key sorting widgets (field list, direction combobox, sort-row configuration). These steps form the minimal causal backbone of the task—"open filter panel → add sort rows → configure fields/directions → apply sort"—and enable the hinter to produce crisp, structured hints such as "expand the filter panel, click 'Add Sort,' choose fields in priority order, select directions, then click 'Run filter' to apply." In contrast, non-zoomed traces generate noisy and unfocused hints like "open Personalize List, add fields, wait for headers to load, then click headers or use Actions ¿ Sort," which do not reliably reflect the canonical work-

flow. This difference explains the $0.1 \rightarrow 0.7$ success jump and why no improvement appears without zooming.

## 6 CONCLUSION

We present DR. HINTER, an agentic system that distills large offline traces into short, retrievable hints that help agents overcome common failure modes. DR. HINTER uses a zooming module to identify critical decision points in long trajectories. A reflection step then distills these segments into reusable strategies and pitfalls. The resulting hints are compact, transparent, and easily injected at inference without fine-tuning. Experiments on MiniWoB++, WorkArena-L1, and WebArena-Lite show improvements over strong baselines, including gains in both out-of-goal and out-of-task generalization. Ablations further highlight how retrieval design, hinter capacity, and the inclusion of failed trajectories shape downstream performance, offering actionable insights for future applications. We view this work as a step toward data-centric adaptation of LLM agents, where past trajectories, documents, and human instructions are systematically mined into reusable knowledge for more robust and resilient decision-making.

**Reproducibility Statement.** The reproducibility of experiments on web agents poses several challenges, as it relies on a software stack for hosting the environment server and the backend of the web agent. To address this, we rely on AgentLab and BrowserGymChezelles et al. (2025), a framework designed for evaluating agents with reproducibility in mind. Among other features, the version of all installed packages used during the experiments is saved in the experiment results. In addition to open-sourcing our code, we will also provide all experiment traces as provided by AgentLab. In the meantime, an anonymized codebase is provided in the supplementary materials.

For the reproducibility of our method, Section 3, which provides a detailed description of the DR. HINTER framework, while Section 4 specifies benchmarks, baselines, and evaluation protocols. Appendix C includes the full prompts used for hint generation and retrieval, Appendix A describe documentation and human hint collection procedures, and Appendix E.2 provides case studies with reasoning traces. All datasets (MiniWoB++, WorkArena-L1, and WebArena-Lite) are publicly available, and we include details of our offline data collection and augmentation pipeline in Section 4.

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

## A    DOCUMENTATION SEARCH AS HINTS FOR LLM AGENTS

We explore the use of documentation search as a hinting mechanism, enabling agents to retrieve relevant knowledge directly from official platform resources. Specifically, we scrape documentation from ServiceNow[2] for WorkArena-L1, and from GitLab[3] and shopping websites[4] for WebArena. Each webpage is converted into a cleaned markdown file with a structured header that records metadata such as the page title, summary, keywords, and breadcrumbs.

**Experimental Setup**    To evaluate how best to retrieve relevant hints, we explore three complementary design dimensions:

- **Retrieval method**. We compare sparse retrieval with BM25 (Robertson et al., 1995) against dense retrieval using pretrained embeddings (Karpukhin et al., 2020).

- **Query formulation**. We test using the raw task goal as the query versus prompting the LLM to generate a more specific query from the current task context. This comparison mirrors episode-level hints versus step-level hints.

- **Granularity of retrieval**. We contrast retrieving full documentation pages with retrieving structured chunks. In the chunked setting, we align snippets with the markdown hierarchy, treating each section as an independent unit without overlap.

Information about the extracted documentation webpages can be found in Table 3.

We evaluate configurations on WorkArena-L1 using GPT-5-mini as the base model. Sparse retrieval is implemented with `bm25s` Lù (2024), while dense retrieval uses `embeddinggemma-300m` Vera et al. (2025) with Faiss Douze et al. (2024). For reformulated queries, GPT-5-mini generates context-aware search strings. To ensure fairness across setups, we fix the retrieval depth: the full-page setting returns the top 3 pages, and the chunked setting returns the top 5 section-level snippets.

Table 4: Comparison of Documentation Search Settings for Web-Browsing Agents

Table 3: Documentation Corpus Statistics: Number of Pages and Chunks per Platform

| Platform | # Pages | # Chunks |
|----------|---------|----------|
| ServiceNow | 60,967 | 287,271 |
| GitLab | 2,654 | 35,470 |
| Shopping | 598 | 4,010 |

| Search Type | Query Type | Document Type | Reward |
|-------------|-----------|---------------|--------|
| N/A | N/A | N/A | 0.61 |
| Sparse | Goal | Full | **0.64** |
| Sparse | Goal | Chunk | 0.63 |
| Sparse | LLM | Full | 0.62 |
| Sparse | LLM | Chunk | 0.63 |
| Dense | Goal | Full | 0.60 |
| Dense | Goal | Chunk | 0.59 |
| Dense | LLM | Full | 0.62 |
| Dense | LLM | Chunk | 0.61 |

**Results**    The ablation results across these configurations are reported in Table 4. Overall, we find that **a simple retrieval framework is highly competitive**. Using BM25 with the task goal as the query and retrieving full pages achieves performance on par with more complex dense retrieval and LLM query reformulation setups. This configuration is also faster and easier to implement, making it a strong baseline for documentation-based hinting. While advanced retrieval pipelines provide only marginal gains, simplicity and efficiency often suffice for supplying LLM agents with actionable documentation hints. Dense retrieval in particular underperforms, likely due to embeddings being less attuned to domain-specific technical terminology.

---

[2]https://www.servicenow.com/docs/

[3]https://docs.gitlab.com/

[4]https://experienceleague.adobe.com/en/docs/commerce-admin/user-guides/home

**Discussion**   In most cases, we find that documentation pages are not a reliable source of instructions for navigating complex user interfaces. Unlike tutorials designed for end-users, documentation rarely specifies how to perform low-level interactions such as clicking, scrolling, or filling forms. As a result, retrieved passages often contain information that is only tangentially related to the task at hand. Encouragingly, the agent is generally able to disregard irrelevant context and maintain a similar level of performance, even if individual successes and failures shift across tasks. In other words, documentation hints can occasionally distract the agent, but the net effect on performance is largely stable when the provided context is unhelpful.

The impersonation task stands out as the most notable case where documentation significantly improves performance. Without hints, GPT-5-mini frequently refuses to act, interpreting "impersonation" as unsafe rather than recognizing it as a legitimate ServiceNow feature. This reflects an alignment artifact, where the model overgeneralizes safety constraints to benign enterprise contexts. Providing the impersonation documentation resolves this issue, enabling successful execution. This example highlights the dual benefit of documentation retrieval: it can both supply missing procedural knowledge and clarify task intent in ways that help override misaligned safety refusals. In contrast, tasks such as filtering and sorting show degradation primarily due to skill-based errors, underscoring that documentation hints are most impactful in cases where alignment conflicts, rather than procedural gaps, are the limiting factor.

**Limitations**   A key limitation of documentation-based hinting is its reliance on the availability of high-quality resources. Within WebArena, only GitLab and Shopping/Shopping Admin tasks are supported by relevant documentation, and even these are far less comprehensive than ServiceNow's materials in WorkArena-L1. Other platforms, such as OpenStreetMap and Postmill, offer little to no user-facing documentation. As also noted by Song et al. (2024), the breadth and quality of documentation directly affect agent performance, particularly for tasks requiring API-level interaction. This underscores that documentation-based approaches may not generalize uniformly across platforms.

## B   HUMAN HINT COLLECTION

To gather high–quality hints from humans, we designed an interactive annotation interface that places the human annotator in the loop of action selection. At each step of a task, the model proposes a list of candidate actions. If the correct action is among them, the annotator simply selects it. Otherwise, the annotator can provide a free–form hint that guides the model toward the desired action. The model then regenerates a new set of candidate actions conditioned on this hint, and the cycle continues until the task is successfully completed. This iterative process ensures that we collect both the final action sequence and, importantly, the intermediate natural language hints produced by humans. fig. 7 presents the labeling UI used to collect human hints.

The hints serve to make explicit the reasoning behind otherwise opaque choices. For example, when filtering a table, annotators often wrote instructions such as: *click on the gridcell that says "– choose field –" to pick Category* or *let's do one condition at a time. Click on "choose field" so that we can select Assigned to*. Similarly, when filling a multi–tab form, annotators specified *to set the assignment group, click on the look up icon*. These hints capture localized decision strategies and offer the model additional guidance beyond raw demonstrations. By collecting such hints alongside trajectories, we create a resource that directly encodes human teaching signals and can be reused to improve model alignment with task–specific interaction patterns.

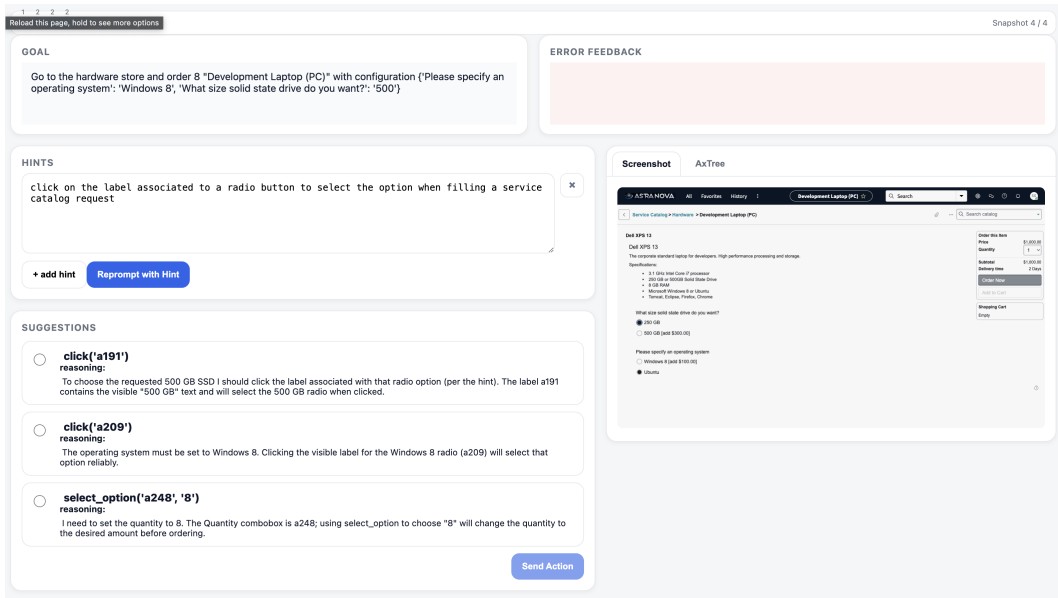

Figure 7: Interactive labeling interface used for human hint collection. Annotators selected actions from a model–generated list, or provided free–form hints when the desired action was missing. The updated candidates were then re–evaluated until the task was completed.

## C  SYSTEM PROMPTS

### C.1  STEP SELECTION

---

**Prompt for Step Selection**

You are a trace summarizer. Given the following execution trace, identify the step or steps that are most important for understanding success or failure. Return the step numbers (starting from 1) and a brief reason why they are important.
=== EXECUTION TRACE ===
Goal: <TASK GOAL>
Step 1: ...
Step 2: ...
=== STEP SELECTION CRITERIA ===
Look for steps that are critical because they:
1. Represent a key decision point or branching moment
2. Show a common mistake that could be avoided
3. Demonstrate a successful strategy or pattern
4. Involve important UI elements or context clues
5. Show timing or sequence dependencies
6. Represent the moment where success/failure was determined
=== STEP SELECTION ===
List the most important step numbers (comma separated) and a brief reason for each.
IMPORTANT: Do not repeat the same step number. Select 1–2 critical steps that provide the most valuable insights for generating actionable hints.
=== THINKING PROCESS ===
Before selecting the most important steps, think through:
1. Which steps represent critical decision points?
2. Which steps show avoidable mistakes?
3. Which steps demonstrate successful strategies?
4. Which steps involve important UI/context clues?
5. Which steps show timing or sequence dependencies?
6. Which steps mark where success/failure was determined?
Think step by step and analyze carefully before making your selection.

---

## C.2 STEP-SEQUENCE HINTING

---

**Prompt for Step-Sequence Hint Generation**

Task: `<TASK NAME>`
**=== STEP SEQUENCE ANALYSIS (¡N consecutive steps) ===**
Goal: `<TASK GOAL>`
Step i: `Observation(s), Agent's reasoning, Action taken, Error encountered, Current reward`
Step i+1: `Observation(s), Agent's reasoning, Action taken, Error encountered, Current reward`
...
=== STEP-SEQUENCE HINT GENERATION ===
Based on the sequence of `<N>` consecutive steps above, provide a concise, actionable hint that explains:
1. hat the agent accomplished across these steps.
2. What the agent should do next based on the context.
3. How to recognize when this sequence is needed.
4. Common mistakes to avoid during this sequence.
=== STEP-SEQUENCE GUIDANCE ===
Focus on:
– What changed in the environment across these steps.
– What the agent learned or accomplished.
– The next logical action.
– How to recognize the right moment for that action.
– The pattern or workflow this sequence represents.
**Include the full Hint Requirements" and Output Format" as in Appendix C.3.**

---

## C.3 HINT GENERATION

---

**Prompt for Hint Generation (Single / Multi-Trace)**

**System role**
You are a hint generation expert. You MUST respond using the structured format with `<think>`, `<topic>`, and `<hint>` tags. Use the `<think>` section for thorough analysis (200–800 words) and the `<hint>` section for concise, actionable guidance (under 256 tokens, single line).
=== INPUT ===
Task: `<TASK NAME>`
Goal: `<TASK GOAL>`
(Optional) Documents/Instructions: `<SHORT SNIPPETS OR NONE>`
**Execution trace(s):**
Step 1: `Observation(s), Agent's reasoning, Action taken, Error encountered, Current reward`
Step 2: `Observation(s), Agent's reasoning, Action taken, Error encountered, Current reward`
...
(Repeat for each provided trace when multiple traces are given)
=== HINT REQUIREMENTS ===
IMPORTANT: Keep your hint SHORT and write it as a SINGLE LINE without line breaks.
Focus on:
– Common pitfalls or errors to avoid
– Specific strategies that work well
– Important details and UI cues to pay attention to
– Step-by-step guidance if multiple actions are required
=== ENHANCED REQUIREMENTS ===

---

**Generalizability**

– Make hints general enough to apply to similar tasks, not just this specific instance.

– DO NOT include: specific usernames, literal task content strings, element IDs like [123], domain-specific secrets.

– DO include: reusable UI patterns (buttons, links, form fields), common workflows, robust strategies.

**Specificity & Actionability**

– Use exact UI text only when it represents common patterns (e.g., button labels like 'Submit').

– Specify element types and positions when relevant (e.g., button at the bottom of the form).

– Provide clear step ordering when multiple actions are needed.

**Structure & Length**

– Hint under 256 tokens, single line, no line breaks.

– Focus on what to do, not why it works.

– Use single quotes (') and *never* double quotes (") in the hint.

**Topic Tag**

– Always provide one short sentence describing the applicability topic inside `<topic>` tags (e.g., `filtering the table`, `multi-tab form filling`).

– If a line `SUMMARIZATION: <summarization>` is present in the input, incorporate it into the `<topic>` description.

Known applicability topics: `<TOPIC LIST IF AVAILABLE>`

=== OUTPUT FORMAT ===

`<think>`

Your reasoning about the traces, patterns, decisive steps, and reusable strategies (200–800 words).

`</think>`

`<topic>`

One short sentence describing the general task topic (e.g., `filtering the table`).

`</topic>`

`<hint>`

A single-line, concise, actionable hint under 256 tokens (use single quotes, no line breaks).

`</hint>`

=== THINKING SECTION GUIDANCE ===

– Analyze the execution traces in detail.

– Identify key patterns, mistakes, and successful strategies.

– Explain (in `<think>`) why certain approaches work or fail.

– Consider multiple perspectives and edge cases.

– Aim for 200–800 words of thoughtful analysis.

=== HINT SECTION GUIDANCE ===

– Focus on the most critical action(s) the agent should take next.

– Avoid lengthy explanations or multiple examples.

– Prioritize what to do; keep it executable.

– Keep it under 256 tokens; use single quotes only.

=== EXAMPLES (GOOD) ===

**Example 1 - Navigation:**

`<think>`

Looking at the execution traces, the agent often fails by using the global search instead of the left-side Application Navigator. Successful runs type into 'Filter/Filter navigator' and click module links after the app expands. Repeated clicks on admin menus are unnecessary; the key is filtering in the left panel and then selecting the specific module entry once visible.

`</think>`

`<hint>`

Use the Application Navigator (left panel) with the 'Filter/Filter navigator' input to find and open modules; do not use the global search bar at the top.

`</hint>`

**Example 2 - Form Submission**

`<think>`

Agents fail when expecting a 'Submit' label; successful runs click whichever action completes the flow ('Save', 'Create', or 'Submit'). Enter does not submit; explicit clicks are required.
```
</think>
<hint>
```
At the bottom of the form, click the action button that completes the flow (e.g., 'Save', 'Create', or 'Submit') instead of pressing Enter.
```
</hint>
```
=== EXAMPLES (BAD) ===
– Click the button with ID [123] to submit the form. (too specific)
– Enter 'john.doe@email.com ' in the email field. (too specific)
– This task requires careful attention to detail. (too vague)
– The agent should understand the context before proceeding. (explanatory, not actionable)
– Click the "Submit" button to continue. (uses double quotes)

## C.4 TWO-TRACE COMPARISON (DESIRED VS. UNDESIRED)

**Prompt for Two-Trace Comparison (Desired vs. Undesired)**

You will be provided with a **desired (successful)** and an **undesired (failed)** trajectory for the same task. Identify the *first* action where they diverge, explain why it leads to success vs. failure, and produce a general, reusable hint.
=== INPUT ===
Task: `<TASK NAME>`
Goal: `<TASK GOAL>`
— **Desired trajectory** —
```
Step 1: Observation(s), Agent's reasoning, Action taken, Error
encountered, Current reward
Step 2: Observation(s), Agent's reasoning, Action taken, Error
encountered, Current reward
```
. . .
— **Undesired trajectory** —
```
Step 1: Observation(s), Agent's reasoning, Action taken, Error
encountered, Current reward
Step 2: Observation(s), Agent's reasoning, Action taken, Error
encountered, Current reward
```
. . .
**SUMMARIZATION:** `<ONE-LINE CONTEXT SUMMARY IF AVAILABLE>`
=== COMPARISON GUIDANCE ===
1. Identify the first differing action and its local context.
2. Explain (in `<think>`) why one path succeeds and the other fails.
3. Derive a general rule that applies beyond this instance; avoid task-specific literals.
4. Follow the successful (desired) trajectory; do not invent steps absent from it.
=== OUTPUT FORMAT ===
```
<think>
```
Analysis of the first divergence, its effect on progress, UI/context cues to detect it, and a reusable rule (200–800 words).
```
</think>
<topic>
```
Short applicability topic (e.g., `using the application navigator vs. global search`).
```
</topic>
<hint>
```
Single-line, general, actionable guidance under 256 tokens; preferably in the form: When ¡status¿, do ¡action¿ or Avoid ¡pitfall¿ and instead ¡action¿. Use single quotes.
```
</hint>
```

## C.5 STEP-ZOOM HINTING

---

**Prompt for Step-Zoom Hint Generation**

Task: `<TASK NAME>`
**=== ZOOMED-IN STEPS ===**
Goal: `<TASK GOAL>`
(For each step in the trace, include:)
Step k: `Observation(s), Agent's reasoning, Action taken, Error encountered, Current reward`
(For each step identified as important, additionally include the most informative structural view, e.g., AXTree or HTML.)
=== HINT GENERATION ===
Based on the most important step(s) above, provide a concise, actionable hint that would help an agent avoid common mistakes and succeed at this task.
=== STEP-FOCUSED GUIDANCE ===
1. Pay special attention to:
2. What makes this step decisive for success/failure.
3. The specific UI elements or context guiding the correct action.
4. Common mistakes at this decision point.
5. How to recognize when this step is needed.
6. The correct sequence or timing for this action.
**Include the full Hint Requirements" and Output Format" as in Appendix C.3.**

---

## C.6 DUAL-TRACE STEP-ZOOM

---

**Prompt for Dual-Trace Step-Zoom Analysis**

Task: `<TASK NAME>`
**=== DUAL TRACE STEP ZOOM ANALYSIS ===**
For each trace (desired and undesired), provide:
– Outcome summary (successful/failed) and Goal.
– Steps with: `Observation(s), Agent's reasoning, Action taken, Error encountered, Current reward`.
– Mark **IMPORTANT STEP** for the selected critical steps and include the relevant structural view (AXTree/HTML) for those steps.
=== DUAL TRACE HINT GENERATION ===
Based on the most important step(s) across both traces, provide a concise, actionable hint that helps avoid the observed failure.
=== DUAL TRACE STEP-FOCUSED GUIDANCE ===
Focus on:
1. Patterns emerging across both traces at critical steps.
2. Differences between correct and incorrect actions at those points.
3. The UI elements or context that disambiguate the right action.
4. Common mistakes at similar decision points.
5. How to recognize when these critical steps are needed.
6. The correct sequence/timing for actions at these points.
**Include the full Hint Requirements" and Output Format" as in Appendix C.3.**

---

## C.7 CONTEXT IDENTIFICATION

---

**Prompt for Context Identification (Pre-Retrieval)**

You are a helpful assistant that identifies the context of a task based on trace information. You will see the prefix of a trajectory up to the first divergence between two traces. Summarize the current status to guide retrieval of relevant hints.

=== INPUT (TRACE PREFIX) ===

GOAL: `<TASK GOAL>`

Step 1: `Observation(s), Agent's reasoning, Action taken, Error encountered, Current reward`

Step 2: `Observation(s), Agent's reasoning, Action taken, Error encountered, Current reward`

. . . (up to the first differing action)

=== INSTRUCTIONS ===

Before choosing an action, query memory/documentation by first generating a brief, general summary of the current status to help identify useful hints.

Return your answer as follows:

`<think>`chain of thought`</think>`

`<context>`one short sentence summary`</context>`

=== EXAMPLE ===

`<think>`

I have to sort by client and country. I could use the built-in sort on each column but I'm not sure if I can sort by both at the same time.

`</think>`

`<context>`

The user is preparing to apply multi-column sorting and needs guidance on adding the next criterion.

`</context>`

---

## D MORE RESULTS

**How much faster is parallelized hint generation?**

The original AutoGuide guideline extraction Fu et al. (2024) module is implemented sequentially, which limits scalability. To demonstrate the efficiency of our approach, we implemented a parallelized version of hint generation that distributes trajectories across multiple workers. As shown in fig. 8, our parallel implementation achieves nearly a $20\times$ speedup over sequential hinting, enabling large-scale hint generation on complex benchmarks. This improvement makes it practical to construct diverse and comprehensive hint databases without prohibitive computational overhead.

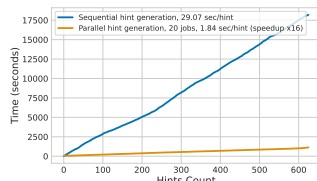

Figure 8: Parallelized hint generation.

# E  HINT ANALYSIS

## E.1  HINT STATS

Table 5: MiniWoB++ Hint Database Statistics by Method, Base Model, and Hinter Model

| Hinter Method | Base Model | Hinter Model | Total Entries | Unique Tasks | Avg Hints/Task |
|---|---|---|---|---|---|
| AutoGuide-v1 | gpt-5-mini | gpt-5 | 117 | 28 | 4.17 |
| | | gpt-5-mini-2025-08-07 | 139 | 28 | 4.96 |
| | gpt-5-nano | gpt-5 | 157 | 36 | 4.38 |
| | | gpt-5-mini-2025-08-07 | 174 | 36 | 4.83 |
| DR. HINTER (w/o zoom) | gpt-5-mini | gpt-5 | 625 | 125 | 5.00 |
| | | gpt-5-mini-2025-08-07 | 614 | 125 | 4.91 |
| | gpt-5-nano | gpt-5 | 625 | 125 | 5.00 |
| | | gpt-5-mini-2025-08-07 | 619 | 125 | 4.95 |
| DR. HINTER | gpt-5-mini | gpt-5 | 625 | 125 | 5.00 |
| | | gpt-5-mini-2025-08-07 | 618 | 125 | 4.94 |
| | gpt-5-nano | gpt-5 | 625 | 125 | 5.00 |
| | | gpt-5-mini-2025-08-07 | 620 | 125 | 4.96 |

Table 6: WorkArena-L1 Hint Database Statistics by Method, Base Model, and Hinter Model

| Hinter Method | Base Model | Hinter Model | Total Entries | Unique Tasks | Avg Hints/Task |
|---|---|---|---|---|---|
| AutoGuide | gpt-5-mini | gpt-5 | 105 | 21 | 5.00 |
| | | gpt-5-mini-2025-08-07 | 105 | 21 | 5.00 |
| | gpt-5-nano | gpt-5 | 155 | 31 | 5.00 |
| | | gpt-5-mini-2025-08-07 | 155 | 31 | 5.00 |
| DR. HINTER(w/o zoom) | gpt-5-mini | gpt-5 | 194 | 33 | 5.88 |
| | | gpt-5-mini-2025-08-07 | 165 | 33 | 5.00 |
| | gpt-5-nano | gpt-5 | 188 | 33 | 5.70 |
| | | gpt-5-mini-2025-08-07 | 168 | 33 | 5.09 |
| DR. HINTER | gpt-5-mini | gpt-5 | 165 | 33 | 5.00 |
| | | gpt-5-mini-2025-08-07 | 164 | 33 | 4.97 |
| | gpt-5-nano | gpt-5 | 165 | 33 | 5.00 |
| | | gpt-5-mini-2025-08-07 | 163 | 33 | 4.94 |

Table 7: WebArena-Lite Hint Database Statistics by Method and Hinter Model

| Hinter Method | Hinter Model | Total Entries | Unique Tasks | Avg Hints/Task |
|---|---|---|---|---|
| AutoGuide | GPT-5 | 130 | 26 | 5 |
| | GPT-5-Mini | 130 | 26 | 5 |
| DR. HINTER | GPT-5 | 825 | 165 | 5 |
| | GPT-5-Mini | 825 | 165 | 5 |

## E.2  HINT EXAMPLES

<think> Both traces involve the same UI pattern: a HTML select element with the multiple attribute, a visible list of options, and a 'Submit' button. The successful trace shows the agent explicitly used modifier-clicks (Control) when selecting multiple items; the agent selected the first target with Control-click, added the second with another Control-click, then clicked 'Submit' and received reward. The failed trace shows a common misunderstanding: the agent used plain clicks when attempting to select a second item, which replaced the first selection rather than adding to it, and clicking 'Submit' then submitted only the last-clicked option and produced no reward. Key patterns: - Presence of multiple attribute on the select control signals multi-selection behavior; this requires modifier keys to accumulate selections. - A plain click on an option typically sets selection to that single option (replaces existing selection). - Using Control (or Command on macOS) while clicking adds or toggles individual options without deselecting others. - Shift-click can be used to select contiguous ranges if needed. Differences between traces: - Successful: used Control-modified clicks for both selections, then clicked 'Submit' — correct sequence. - Failed: used plain clicks resulting in the second click overwriting the first selection, then clicked 'Submit' — sequence wrong because of missing modifier. Specific UI/context cues: - The select element shows multiple="" which is the explicit indicator that multiple selections are allowed. - The button labeled 'Submit' is the final action to apply the selection; ensure desired options are already selected before clicking it. - Sometimes DOM snapshots show the select's value reflecting only one option even when multiple are visually selected; don't rely only on the single 'value' attribute in snapshots — use the correct interaction (modifier-clicks) to ensure multiple options are selected. Common mistakes to avoid: - Forgetting to hold the appropriate modifier key when selecting additional options. - Assuming the second plain click will add to the first selection rather than replace it. - Clicking 'Submit' before verifying all required options have been added. Action sequence to generalize: 1) Identify the select has multiple selection capability. 2) Use Control (or Command) modified clicks to add each non-contiguous option, or Shift to select a contiguous range. 3) After all target options are highlighted, click the 'Submit' button to finalize. This guidance applies broadly to tasks that require selecting several items from a multi-select list and then submitting the form. </think> <topic> selecting an option and clicking the Submit button </topic> <hint> For multi-select lists, hold Control (or Command on macOS) and click each option to add them to the selection, then click the 'Submit' button; avoid plain clicks that replace the current selection and use Shift for contiguous ranges. </hint>

Figure 9: miniwob.click-scroll-list. DR. HINTER with gpt-5-mini as the hinter model and gpt-5-nano as the base model. The performance is improved from 0.6 to 1 on this task after applying hint.

<think> Both traces aim to navigate to the Interaction app's Active module, but they diverge on which navigation context and input to use. The successful trace opens the primary Application Navigator by clicking the All menu, uses the left-panel filter textbox ('Enter search term to filter All menu'), types the app name, waits for the Interaction application to appear and expand, then clicks the 'Active' module link. The failed trace repeatedly operates in the More menus / Workspaces context: it opens More menus, types into the Workspaces filter, and repeatedly clicks the 'Active' link there without getting the expected outcome. Key patterns: (1) using the correct navigator context (All vs Workspaces) matters; the same app and module labels can exist in different menus but only one leads to the intended module in this workflow, (2) typing into an already-focused filter and waiting for results to populate before clicking is necessary, and (3) repeatedly clicking without confirming the proper menu or waiting for navigation to complete is ineffective. Common mistakes observed: choosing the wrong menu context (Workspaces or global search) instead of the Application Navigator, typing into the wrong filter box, and not confirming the Appearance/expansion of the application entry before clicking the module link. The failed trace also shows excessive repeated clicks on the same link, suggesting the agent was unsure whether the right context was selected or whether the click had any effect. Timing/sequence issues: first ensure the desired menu (All / Application Navigator) is expanded and the left-panel filter textbox is focused; second type the app name; third wait for the app entry to expand and its module list to render; fourth click the specific module link. From a generalizable perspective for ServiceNow navigation tasks: always pick the navigation area designed for module access (left Application Navigator), use its dedicated filter input, verify the application container expands and lists modules, then select the module link; avoid using top/global search fields or alternate menu contexts unless specifically required. Also, visually confirm the filter input's placeholder text or the Application Navigator header to ensure you're in the right panel; if results don't appear, clear the filter and retry or re-open the All menu. These steps reduce errors caused by similar labels across different navigation menus and avoid wasted repeated clicking. </think> <topic> navigating modules via the Application Navigator (left panel) </topic> <hint> Open the Application Navigator 'All' menu (left panel), type the application name into the 'Enter search term to filter All menu' textbox, wait for the app to expand and show modules, then click the module link 'Active' — do not use the top/global search or the 'Workspaces' filter. </hint>

Figure 10: workarena.servicenow.all-menu. DR. HINTER with gpt-5-mini as the hinter model and gpt-5-nano as the base model. The performance is improved from 0 to 1 on this task after applying hint.

> **\<think\>** I will open the Filters panel so I can use the Bill-to Name filter to get exact counts per customer (recommended in the hints). Opening Filters is the next step before applying a Bill-to Name query. **\</think\>**
> **\<task_hint\>** Relevant hint: Open Sales > Orders, click 'Filters', set 'Status' to 'Canceled', click 'Apply Filters'; clear the keyword search box, then sort the 'Bill-to Name' column to group names and scan/paginate for the largest group; to verify counts, use the 'Bill-to Name' filter and read 'records found', removing that chip before testing another. **\</task_hint\>** **\<action\>** click('845') **\</action\>**

Figure 11: webarenalite.288. DR. HINTER with gpt-5 as the hinter model and gpt-5-mini as the base model. The performance is improved from 0 to 1 on this task after applying hint.

## F  HINT GENERATION AND RETRIEVAL ALGORITHM

---

**Algorithm 1 DRH: Hint Generation (Zoom & Reflect)**

---

**Require:** Offline trajectories $\{\tau_i\}_{i=1}^N$ (with $x, z, a, r$), optional documents/instructions, summarizer $\mathcal{S}$, hinter $\mathcal{H}$
**Ensure:** Hint database $\mathcal{D}_\mathcal{H} = \{(c, h)\}$
1: **for each** selection of evidence $E \in \{\text{single } \tau, \text{pair } (\tau^+, \tau^-), \text{multi-trace } \{\tau^j\}_{j \in S}\}$ **do**
2:     $c \leftarrow \mathcal{S}(E)$                                            ▷ semantic key / context used for retrieval
3:     **if zooming then**
4:         choose critical step(s) $t^*$ and window $\Delta$
5:         $P \leftarrow P_\tau^{\text{zoom}} = \{z, a, r\}_{1:T} \cup \{x\}_{t^*:t^*+\Delta}$
6:     **else**
7:         $P \leftarrow P_\tau^{\text{full}} = \{x, z, a, r\}_{1:T}$
8:     **end if**
9:     **if contrastive then**
10:        $P \leftarrow$ contrastive prompt built from $(\tau^+, \tau^-)$
11:     **end if**
12:     $h \leftarrow \mathcal{H}(c, P)$                                  ▷ natural–language hint linked to its source
13:     $\mathcal{D}_\mathcal{H} \leftarrow \mathcal{D}_\mathcal{H} \cup \{(c, h)\}$
14: **end for**
15: **return** $\mathcal{D}_\mathcal{H}$

---

**Algorithm 2 DRH: Retrieve & Act**

---

**Require:** Policy $\pi$, database $\mathcal{D}_\mathcal{H} = \{(c, h)\}$, retriever $\rho$, summarizer $\mathcal{S}$, goal $g$, mode $\in \{\text{EPISODE}, \text{STEP}\}$
1: **if** mode $=$ EPISODE **then**                           ▷ goal-conditioned (episode-level) retrieval
2:     $\{h^1, \ldots, h^k\} \leftarrow \rho(g, \mathcal{D}_\mathcal{H})$
3: **end if**
4: **for** $t = 1, \ldots, T$ **do**
5:     Observe $x_t$ and update $\tau_{:t}$
6:     **if** mode $=$ STEP **then**                         ▷ contextual (step-level) retrieval
7:         $c_t \leftarrow \mathcal{S}(\tau_{:t})$
8:         $\{h_t^1, \ldots, h_t^k\} \leftarrow \rho(c_t, \mathcal{D}_\mathcal{H})$
9:         $a_t \sim \pi\big(x_{0:t}, \{h_t^1, \ldots, h_t^k\}\big)$
10:     **else**                                                 ▷ EPISODE
11:         $a_t \sim \pi\big(x_{0:t}, \{h^1, \ldots, h^k\}\big)$
12:     **end if**
13:     Execute $a_t$, receive $(x_{t+1}, r_t)$
14: **end for**
15: **return** $\{a_t\}_{t=1}^T$

---

