# OpenReview forum: "Deep Reflection Hinting: Leveraging Offline Knowledge for Improving LLM Agents Adaptation"
_ICLR.cc/2026/Conference — Submitted to ICLR 2026_

### Official Review · Reviewer_iAEN · 2025-10-27

**Soundness:** 2
**Presentation:** 3
**Contribution:** 3
**Rating:** 6
**Confidence:** 4

**Summary:**

The paper introduces DR. HINTER, an agentic framework that turns offline trajectories—both successful and failed—into compact, reusable natural-language hints to improve LLM agents without fine-tuning. A zooming module identifies critical decision points within long traces, and a reflection step distills these segments into strategy and pitfall hints, each indexed by a semantic key for retrieval. Hints can be generated from single traces, contrastive pairs, or multi-trace aggregates, enabling coverage even when only failures exist, and are retrieved at inference either step-by-step (contextual) or once per episode (goal-conditioned). This approach is benchmark-independent, scalable, and transparent, and it supports closed-source models by shifting computation offline to a more capable “hinter” model. Across MiniWoB++, WorkArena-L1, and WebArena-Lite, DR. HINTER consistently outperforms ReAct and an AutoGuide reimplementation, sustains competitive out-of-task generalization, and proves more practical than documentation or human-authored hints. Ablations show that zooming improves hint quality and larger hinter models yield further gains, especially on long-horizon tasks.

**Strengths:**

1. The general idea to distill off-line traces into compact hints looks promising.
2. Some designs like zooming and extracting hints from both successful and failure trajectories sound practical
3. The writing is clear in general.

**Weaknesses:**

1. I think the experimental setting is not very good. Based on the descriptions from line 252-275, the authors seems to collect trajectories from benchmark data. Even in out-of-task generalization, the hints from other tasks on the same datasets are used, which means they use hints of similar tasks in the same distribution in the evaluation. In WebArena, the benchmark uses task templates to construct tasks, where multiple tasks only differ in a few values, while the trajectories to complete them are very similar. I think this gives the question about whether the results are overfitting and how the proposed methods can be applied in general tasks.
2. The experiments remain the GPT5--nano and GPT5-mini, but the effectiveness on more powerful models like GPT-5 or Claude-4.1 remain unclear.
3. From example in Appendix E.2, I feel that the generated hints are specific to certain tasks. It would be better to further abstract hints or scale the number of hints to make this mechanism generalizable.

**Questions:**

See above

---

> ### Author Response · Authors · 2025-11-20
> **Response**
>
> We thank the reviewer for their careful reading, acknowledging the promising general idea, practical design choices like zooming and failure trace extraction, and clarity of writing. We address the concerns regarding the experimental setting and model generalization below.
>
> ---
>
> **Experimental Setting**
>
> We thank the reviewer for raising this point. We should clarify that our experiment setup is designed to evaluate test-time adaptation of LLM agents and we followed the standard evaluation protocols used by prior work [1, 2]. In particular, we evaluate two levels of generalization across all benchmarks (MiniWoB++, WorkArena, WebArena): **in-task** and **out-of-task generalization**. We agree that in-task generalization (such as the reviewer’s example on WebArena) is a relatively weak form of generalization. However, one of the goals of this work is **in-domain adaptation**, helping an agent quickly learn to master a single web environment (e.g.,WorkArena) by reflecting on its past attempts within that environment. This is a crucial requirement for real-world agent deployment. On the other hand, out-of-task generalization, which we evaluate in Section 4.2, follows the cross-template protocol used in prior work [1]. To clarify, each task in WebArena-Lite is instantiated from a **distinct template and requires different action sequences**; removing the source task from the retrieval pool tests whether the system can generalize beyond the specific template used to generate the hints. Our results show that DR-Hinter maintains consistent gains under this setting, indicating that it does not overfit to particular tasks or templates. We clarify the generalization scope of DR.Hinter in the revised manuscript.
>
> An even more challenging generalization level would be cross-category generalization (e.g., GitLab to Reddit in WebArena or filter to sort tasks in WorkArena). We are happy to run such an experiment if the reviewer believes it would meaningfully strengthen the revised submission.
>
> [1] Wang et al, Agent Workflow Memory
>
> [2] Fu et al, AutoGuide: Automated Generation and Selection of Context-Aware Guidelines for Large Language Model Agents
>
> ---
> **Effectiveness on more powerful models**
>
> We thank the reviewer for the suggestion to improve our submission. We provide results using both stronger base agents and stronger Hinter models. We have added new experiments where GPT-5 is used as both the base ReAct agent and the Hinter. We also include results using a vision-based CUA agent with Claude-4.5-Sonnet serving as both the base and the Hinter.
>
> | Base Model / Hinter Model              | Benchmark      | ReAct | DR.Hinter |
> |----------------------------------------|----------------|-------|-----------|
> | GPT-5 / GPT-5                          | MiniWoB++      | 0.715 | 0.739     |
> | GPT-5 / GPT-5                          | WorkArena-L1   | 0.661 | 0.770     |
> | Claude-4.5-Sonnet CUA (vision-based)   | WorkArena-L1   | 0.206 | 0.400     |
> | Claude-4.5-Sonnet CUA (vision-based)   | WebArena-Lite  | 0.273 | 0.291     |
>
>
> These results confirm that DR. Hinter continues to provide **consistent and substantial improvements** even with state-of-the-art models. The gain is due to the method's ability to effectively filter and distill actionable knowledge from **large, noisy environmental context**, which remains a core challenge regardless of the LLM's raw reasoning power.
>
> ---
>
> **Abstracting Hints**
>
> Thank you for raising this. We now included a Curator module that examines all hints associated with a task and (i) merges duplicates via semantic summarization and (ii) generalizes overlapping hints into a single, more robust instruction.
> In our experiments, the curator did not produce large performance gains. We believe this is because our base agent already performs step-level filtering, selecting only the most relevant hint at each decision point, which naturally suppresses noisy or redundant hints. Nonetheless, the Curator improves consistency of the hint set and helps avoid edge-case contradictions, and we report its behavior and results in the updated manuscript.
>
> ---
>
> We are happy to incorporate all changes and hope our responses address the reviewers' concerns, leading to an improved final assessment.

---

> ### Comment · Reviewer_iAEN · 2025-11-25
> **Thank you for the response**
>
> Thanks a lot for the author reply! I decide to maintain my original rating.

---

### Official Review · Reviewer_YdbX · 2025-10-30

**Soundness:** 2
**Presentation:** 2
**Contribution:** 2
**Rating:** 2
**Confidence:** 5

**Summary:**

This paper introduces Deep Reflection Hinter (DR.HINTER), an agentic system that distills offline trajectories into explicit, context-aware hints. DR.HINTER is capable of extracting hints from both successful and failed trajectories simultaneously. Experiments on MiniWoB++, WorkArena-L1, and WebArena-Lite show that DR.HINTER's method outperforms ReAct and AutoGuide, demonstrating the high efficiency of the proposed approach.

**Strengths:**

* The proposed method has strong generality. The author consider the scenario in practical applications where only failed trajectories might be available.

* The proposed method has strong interpretability. The Case Study in Section 5.4 makes it easy to understand the reasons for the method's effectiveness.

* The paper provides a detailed description of the method's design and prompt design, which facilitates reproducibility.

**Weaknesses:**

* A key innovation of this work is the ability to utilize both successful and failed trajectories. However, some existing works also emphasize the ability to extract experience from failed trajectories to guide subsequent inference, such as Automanual [1]. This paper lacks a comparison and experimental contrast with these works.

* Another innovation of this work is the design of the zooming mechanism. However, the paper only proves the importance of this step through ablation experiments and lacks an interpretability analysis or qualitative analysis of this step. For example, it is unclear whether the zooming mechanism would still provide benefits if using large models that support long contexts, such as GPT-5, as the complete trajectory could also provide richer information.

* The paper only verifies that the method achieves performance improvements on GPT-5-Nano and GPT-5-Mini. It lacks experimental analysis on larger models like GPT-5, Gemini 2.5 Pro, and state-of-the-art open-source models like Qwen3-8B.

* A significant advantage of improving model performance through context is strong generalization. However, this paper conducts limited generalization experiments. For instance, the paper should experiment with whether hints obtained on MiniWob++ can be applied to other benchmarks of the same type.

[1] Minghao Chen, Yihang Li, et.al. "AutoManual: Constructing Instruction Manuals by LLM Agents via Interactive Environmental Learning"

**Questions:**

See Weaknesses.

---

> ### Author Response · Authors · 2025-11-19
> **Response (1/2)**
>
> We thank the reviewer for the positive feedback on the generality, interpretability, and reproducibility of our work. We address each concern below.
>
> ---
>
> **Comparison to AutoManual**
>
> We thank the reviewer for pointing out the relevant prior work AutoManual [1]. We acknowledge that we overlooked this paper during our initial literature review and have now incorporated it into the revised manuscript.
>
> We clarify that DR-Hinter differs in both setting and scope. AutoManual operates in an online regime, where the same planner agent continuously revises rules on the same task. DR-Hinter instead works fully offline and can generate hints from any dataset collected by other agents and related tasks, enabling cross-agent and cross-task reuse. AutoManual additionally builds separate skill and reflection libraries through a multi-step online process, whereas DR-Hinter uses a single offline pipeline to directly synthesize natural-language hints suitable for retrieval at test time.
>
> Experimentally, [1] reports results on an easy 53-task subset of MiniWoB++, while our evaluation covers all 125 tasks as well as WorkArena and WebArena-Lite, which present significantly more complex interfaces and failure modes. When we follow the evaluation setup of [1] and evaluate on the same 53-task subset using a minimal vision-based CUA agent, DR-Hinter matches the 98% success rate reported by [1]. This shows that DR-Hinter remains competitive in the narrower domain where AutoManual operates, while scaling to larger and more realistic environments.
>
> [1] Minghao Chen, Yihang Li, et.al. "AutoManual: Constructing Instruction Manuals by LLM Agents via Interactive Environmental Learning"
>
> ---
> **Analysis of the zooming mechanism**
>
> We agree that both qualitative and quantitative evidence strengthen the zooming analysis, and we have added the following case study along with new ablation studies to the revised manuscript.
>
> * **Qualitative analysis:** Case Study: workarena.servicenow.sort-hardware-list. This task shows the strongest impact of zooming: DR-Hinter reaches only 10% success without zooming, but rises to 70% when zooming is enabled, even though the base agent is identical. The gain comes entirely from higher-quality hints generated when the hinter is given zoomed AXTree snapshots. Zooming surfaced four decisive steps drawn from both a successful trajectory (Steps 4 and 8) and a failed one (Steps 14 and 16). Step 4 captures the moment the agent selects the primary sort field and exposes the full dropdown; Step 8 contains a clean post-action AXTree after applying the sort; and Steps 14 and 16, although from a failing run, reveal the same key sorting widgets (field list, direction combobox, sort-row configuration). These steps form the minimal causal backbone of the task, “open filter panel → add sort rows → configure fields/directions → apply sort”, and enable the hinter to produce crisp, structured hints such as *“expand the filter panel, click ‘Add Sort,’ choose fields in priority order, select directions, then click ‘Run filter’ to apply.”* In contrast, non-zoomed traces generate noisy and unfocused hints like *“open Personalize List, add fields, wait for headers to load, then click headers or use Actions > Sort,”* which do not reliably reflect the canonical workflow. This difference explains the 0.1 → 0.7 success jump and why no improvement appears without zooming. We included this qualitative analysis in the paper. We also note that GPT-5 and GPT-5-mini share similar prompt-length constraints when including AXTree snapshots; full traces do not fit, making zooming necessary regardless of model size.
>
> * **Quantitative Analysis:** To isolate the effect of the zooming mechanism, we evaluated an alternative version of DR.Hinter that receives the entire execution trace, including all observations, actions, and think tokens, without any zooming or step selection. Using GPT-5 as both the base agent and the hinter, we find that providing a full single trace yields only marginal gains over the ReAct baseline. In contrast, DR.Hinter with two traces and zooming delivers substantially larger improvements on both benchmarks (see table below). This confirms that the hinter benefits not from having more tokens, but from having the right tokens: comparing trajectories and surfacing the critical AXTree snapshots is essential for producing high-quality, actionable hints.
>
> | Model Pair    | Method                     | MiniWoB++ | WorkArena-L1 |
> |---------------|-----------------------------|-----------|---------------|
> | GPT-5 / GPT-5 | ReAct                      | 0.715     | 0.661         |
> | GPT-5 / GPT-5 | DR.Hinter (full trace)     | 0.718     | 0.715         |
> | GPT-5 / GPT-5 | DR.Hinter (zoom, 2 traces) | 0.739     | 0.770         |

---

> > ### Author Response · Authors · 2025-11-20
> > **Response (2/2)**
> >
> > **Larger base models**
> >
> > We thank the reviewer for the suggestion to improve our submission. We provide results using both stronger base agents and stronger Hinter models. We have added new experiments where GPT-5 is used as both the base ReAct agent and the Hinter. We also include results using a vision-based CUA agent with Claude-4.5-Sonnet serving as both the base and the Hinter. In both settings, DR-Hinter continues to yield consistent improvements, confirming that hint-based adaptation remains valuable even with state-of-the-art models.
> >
> > | Base Model / Hinter Model              | Benchmark      | ReAct | DR.Hinter |
> > |----------------------------------------|----------------|-------|-----------|
> > | GPT-5 / GPT-5                          | MiniWoB++      | 0.715 | 0.739     |
> > | GPT-5 / GPT-5                          | WorkArena-L1   | 0.661 | 0.770     |
> > | Claude-4.5-Sonnet CUA (vision-based)   | WorkArena-L1   | 0.206 | 0.400     |
> > | Claude-4.5-Sonnet CUA (vision-based)   | WebArena-Lite  | 0.273 | 0.291     |
> >
> > ---
> >
> > **Broader generalization experiments**
> >
> > We agree that generalization is an important capability for memory-augmented LLM agents. DR-Hinter already evaluates out-of-task generalization within each benchmark by removing the source task from the retrieval pool, and in these settings the method still provides clear gains, showing that the extracted hints are not tied solely to the exact trajectories from which they were derived.
> > Generalization across different benchmarks (e.g., MiniWoB++, WorkArena, or WebArena) is less realistic within existing benchmark suites. At the moment, it is hard to imagine what kind of hint/knowledge could be transferred. Could you share some ideas if you have any? Perhaps some common widget or some common workflow, but these do not exist in significant quantities across current benchmarks. We believe that this could be achieved if we significantly scale up the number of benchmarks/domains to, e.g., 20, and study generalization in a leave-one-out fashion. That said, we appreciate the motivation behind the suggestion: understanding whether DR-Hinter captures any domain-agnostic signal. One way to explore this would be to evaluate mixed or cross-benchmark hint pools to quantify how environment shifts affect retrieval quality, and we are happy to run such an experiment if the reviewer believes it would meaningfully strengthen the submission.
> >
> > ---
> > We appreciate the reviewer’s thoughtful feedback. DR. Hinter’s central contribution is a unified framework for extracting actionable guidance from imperfect offline traces, made possible by our zooming mechanism, which isolates high-value decision points in long-horizon web tasks with large, noisy AXTrees. Zooming is essential for producing high-quality hints and enables substantial performance gains even for stronger base models.
> >
> > We are happy to incorporate any additional analyses or clarifications that would assist the reviewer in reassessing their score.

---

> > > ### Comment · Reviewer_YdbX · 2025-11-26
> > >
> > > Thanks for the detailed responses, which addressed most of my concerns. Consdering the contribution and impact on the field, I decide to raise my rating to 4.

---

> > > > ### Author Response · Authors · 2025-11-28
> > > > **Follow-up**
> > > >
> > > > Thank you for your engagement and for raising your score. Your feedback has directly helped us improve the submission. We appreciate your positive assessment and your acknowledgment that our rebuttal addressed most concerns. If there are any remaining concerns preventing a higher assessment, we would be glad to clarify or strengthen those points while the discussion window is still open.

---

### Official Review · Reviewer_bPLF · 2025-11-01

**Soundness:** 3
**Presentation:** 3
**Contribution:** 2
**Rating:** 2
**Confidence:** 3

**Summary:**

The paper proposes Deep Reflection Hinting (DR. HINTER), an offline pipeline that distills agent trajectories—successful and unsuccessful—into natural-language hints keyed by semantic summaries for retrieval at inference time. The pipeline has three main stages: (i) Zoom & Reflect, which identifies critical steps within long traces and prompts an LLM (“hinter”) to produce step-focused guidance; (ii) Hint indexing via semantic keys; and (iii) Retrieve & Act, which injects hints either once per episode (goal-conditioned) or per step (context-conditioned). Experiments on MiniWoB++, WorkArena-L1, and WebArena-Lite report gains over ReAct and AutoGuide; ablations assess out-of-task generalization, the effect of larger hinters, and documentation/human-hint baselines.

**Strengths:**

* The paper provides a clear, modular pipeline with ablations to justify some choices (e.g., LLM “zooming”), and the episode- vs. step-level retrieval modes are well-motivated.
* The method leverages failed trajectories to mine “what not to do,” enabling hint creation even when no success trace exists—useful in low-success regimes.
* Broad evaluation across three web-agent benchmarks, including documentation and human-hint comparisons; out-of-task results and hinter-scaling ablations further add empirical soundness.
* Qualitative case studies in the appendix support interpretability claims.

**Weaknesses:**

My main concern is the novelty aspect, but I am open to revising my assessment during the discussion period.
* Conceptual novelty feels incremental relative to prior works. DR. HINTER’s main advances— zooming, single/failed/multi-trace, etc.—read as engineering refinements. Additionally, there are relevant prior works that are not discussed (e.g. \[1\]).
* **Most components rely on prompts to an LLM.** The paper should be more explicit about exactly which roles are handled by LLM prompting (e.g., the step selection) and which are not.
* **Domain scope.** All experiments are browser-based; it’s unclear how the hint format transfers to other settings. Such empirical specificity should be explicit in the title or be accompanied by an additional domain.

\[1\] Holt, S., Luyten, M. R., & Pouplin, T. (2025). Improving LLM Agent Planning with In-Context Learning via Atomic Fact Augmentation and Lookahead Search. *ArXiv*. https://arxiv.org/abs/2506.09171

**Questions:**

1. How do you address duplicate, contradictory, or incorrect hints?
2. Will you release the AutoGuide reimplementation so others can reproduce results?

---

> ### Author Response · Authors · 2025-11-19
> **Rebuttal response**
>
> Thank you for your review. We appreciate that you found our work effective at leveraging failed trajectories in low-success regimes, comprehensive in evaluation with proper ablations across benchmarks, and interpretable through qualitative case studies. We are encouraged by your openness to revising your assessment. We address each concern below:
>
> ---
>
>  **Novelty and Prior Work**
>
> Thank you for pointing us to [1], which we have added to our revision. This work appeared on arXiv in June 2025, shortly before our submission, making it concurrent rather than prior work. We believe this timing provides important context for assessing the value of our work.
>
> Beyond timing, the two methods address fundamentally different challenges. [1] operates on tasks with short observations (approximately 100–200 tokens in ALFWorld) that fit comfortably within context windows, and it focuses on planning augmentation via lookahead search using atomic facts. DR-Hinter tackles web tasks with extremely large accessibility tree observations (10K–50K tokens per page). As observation size grows, zooming becomes essential (Refer to the General Response for added ablation experiments). Without it, even a single trajectory exceeds the context window, making multi-trajectory processing impossible.
>
> In [1], atomic facts represent minimal symbolic world-state information for search, while DR-Hinter’s hints are natural-language abstractions that encode strategies and pitfalls from multiple trajectories. Because hints guide decision-making rather than reconstructing state, they remain effective even in large, noisy web environments.
>
> DR-Hinter also introduces additional contributions, including synthesizing hints by comparing multiple trajectories simultaneously (+7% on WorkArena, +3% on MiniWoB over single-trace without zooming) and extracting actionable guidance from failed traces in zero-success regimes. These capabilities are specifically required for complex, web-scale tasks, as our experiments show.
> We see the methods as complementary, addressing different points along the agent-complexity spectrum.
>
> [1] Holt, S., Luyten, M. R., & Pouplin, T. (2025). Improving LLM Agent Planning with In-Context Learning via Atomic Fact Augmentation and Lookahead Search. ArXiv.
>
> ---
>
> **Domain Scope**
>
> We agree that domain scope should be explicit and will modify the title to: “**Deep Reflection Hinting: Leveraging Offline Knowledge for Improving Web Agents Adaptation**”.
>
> We believe Dr. Hinter's approach can be generalized to other long-horizon decision making domains, but we chose browser tasks as our primary testbed because they represent real-world agentic deployment scenarios with complex action spaces, long horizons, and natural failure prevalence, making them ideal for evaluating hint generation from imperfect demonstrations.
>
> ---
>
> **Clarification of LLM vs. Non-LLM Components**
>
> We agree that the distinction between LLM-based and non-LLM components should be made more explicit. In the revision, we clarifed this directly in the section 3 and in Figure 2.
>
> * **LLM-based modules**: The zooming module, summarizer, hinter, and retriever are all implemented as LLM calls (Sec. 3.2–3.3; prompts in App. C). The base ReAct policy is also an LLM that consumes retrieved hints at inference.
> * **Non-LLM components**: The hint database and its indexing, as well as embedding vector matching, are non-LLM operations, along with the lightweight storage/orchestration logic used during retrieval.
>
> ---
>
> **Q1. How do you address duplicate, contradictory, or incorrect hints?**
>
> Thank you for raising this point. We tested including a curator module that examines all hints associated with a task and removes lexical duplicates, merges semantically similar hints through summarization, and generalizes overlapping hints into a single, more robust instruction. In practice, this curation step provides diminishing returns and does not lead to large performance changes in our experiments. This is partly due to our agent design, which already includes a hint-summarization step before taking an action, naturally suppressing noisy, redundant, or partially contradictory hints. Importantly, we rarely observe any tasks where incorporating hints led to performance degradation. Nonetheless, the Curator improves the overall consistency of the hint set and helps avoid edge-case contradictions, and we report its behavior and results in the updated manuscript.
>
> ---
>
> **Q2: Code Release**
>
> Yes. Upon acceptance we will release: the full DR. Hinter implementation, our AutoGuide reimplementation, all generated hint datasets, and evaluation scripts. This ensures full reproducibility for all benchmarks.
>
> ---
>
> We thank the reviewer for their thoughtful feedback. We are happy to incorporate any further changes that would help the reviewer reassess their score and would welcome guidance on which updates would be most useful during the discussion period.

---

> > ### Author Response · Authors · 2025-11-28
> > **Follow-up**
> >
> > Thank you again for the thoughtful review. We hope our rebuttal helps clarify your concerns, and we would greatly appreciate the opportunity to engage and strengthen the submission before the discussion period closes.

---

### Author Response · Authors · 2025-11-20
**General response**

We thank the reviewers for their constructive feedback. Reviewers highlighted several strengths of our work, including **comprehensive evaluation (bPLF), leveraging both success and failures (bPLF),  interpretability (bPLF, YdbX), generality (YdbX), reproducibility (YdbX), practical design (iAEN)**, and **clarity of writing (iAEN)**. Below, we summarize the common concerns raised across the reviews and describe the major additions made to the revised manuscript. We also provide detailed, point-by-point responses to each reviewer. The main additions are:

**1. Zooming: qualitative + quantitative analysis.**

As requested by **YdbX**, we added a detailed case study showing how zooming isolates the decisive AXTree snapshots that define a task’s causal backbone, enabling high-quality hint synthesis (e.g., 0.1→0.7 improvement on workarena.servicenow.sort-hardware-list). We also include new quantitative comparisons demonstrating that providing the **full trace** to the hinter yields only marginal gains (MiniWoB++: 0.715→0.718; WorkArena-L1: 0.661→0.715), whereas **zooming with two traces yields much larger improvements** (MiniWoB++: 0.739; WorkArena-L1: 0.770). This confirms that the hinter benefits from selecting critical steps, not simply from adding more tokens.


**2. Stronger base and hinter models.**

 To address concerns about generality to stronger base models (**YdbX, iAEN**), we added results using **GPT-5** both as the base ReAct agent and as the hinter. We also tested a vision-based CUA agent with **Claude-4.5-Sonnet** serving as both the base and the Hinter model. In all cases across MiniWoB++, WorkArena-L1, and WebArena-Lite, DR. Hinter continues to provide consistent and often substantial gains (e.g., on WorkArena-L1: GPT-5 0.661 → 0.770; on WebArena-Lite: CUA-Claude-4.5-Sonnet 0.206 → 0.40).

**3. Curator module for hint abstraction.**

We tested adding a curator module that merges semantically similar hints (**bPLF, iAEN**), and generalizes overlapping hints into a cleaner instruction set. While the curator has a modest impact on final performance, largely because our agent already performs step-level hint filtering, it improves consistency, avoids redundant or contradictory hints.

---

We also added clarifications and explanations for our experimental setup and generalization scope (**bPLF, YdbX, iAEN**), comparison to existing works (**bPLF, YdbX**), and framework details (**bPLF**).
All of these additions, including new analyses, ablations, experiments, and discussions, are now incorporated into the revised manuscript to improve clarity and completeness, further strengthening the quality and contribution of the work.

---

### Author Response · Authors · 2025-12-03
**Summary of the Rebuttal phase**

Dear ACs, and SACs,

This note summarizes the reviews, our rebuttal responses, and the discussion. Reviewers highlighted several strengths of the submission, including **comprehensive evaluation (bPLF), leveraging both success and failure data (bPLF), interpretability (bPLF, YdbX), generality (YdbX), reproducibility (YdbX), practical design (iAEN), and clarity of writing (iAEN)**. Below we summarize the major concerns and our responses.

- ****Novelty and Relation to Prior Work (bPLF, YdbX)**** Reviewers questioned whether DR.Hinter’s contributions were sufficiently novel and requested comparison to [1] and [2]. We added explicit comparisons. We clarified that **[2] operates in short-context environments where full trajectories fit into context, while DR.Hinter targets large web environments with 10k to 50k token AXTrees that require selective extraction through zooming.** We also differentiated DR.Hinter from [1], which is an online rule-construction system tied to a single agent and single-task progression. DR.Hinter is fully offline, supports cross-agent and cross-task reuse, and produces retrieval-ready hints. We added controlled experiments on the same 53-task MiniWoB++ subset used by [1], showing that DR.Hinter matches its performance while scaling to more complex environments. Reviewer YdbX found these clarifications sufficient and increased their score to 4.

- ****Zooming Mechanism: Qualitative and Quantitative Evidence (YdbX, bPLF)**** Reviewers requested clearer evidence for why zooming is essential and whether large models could instead process full traces. We added a detailed case study on workarena.servicenow.sort-hardware-list showing that **zooming isolates the causal backbone of the task and increases accuracy from 0.1 to 0.7 with no change to the base agent.** Without zooming, hints become diffuse and unreliable, confirming that zooming improves hint quality rather than simply adding more tokens. We also added a full-trace ablation with GPT-5 as both base agent and hinter. Full-trace prompting yields only small improvements (MiniWoB++ 0.715 to 0.718 and WorkArena-L1 0.661 to 0.715), while zooming with two traces provides substantially larger gains (0.739 and 0.770). This shows that zooming is necessary even for high-capacity models.

- ****Experimental Setting, Generalization Scope, and Potential Overfitting (iAEN, YdbX)**** Reviewers raised concerns about template-level overfitting and the scope of generalization. We clarified that our evaluation follows established protocols and measures both in-task and out-of-task generalization. Out-of-task generalization removes the source task from the retrieval pool and aligns with cross-template evaluation used in prior work. **DR.Hinter maintains consistent gains under this setting, indicating that it does not rely on template-specific artifacts.** We acknowledged that broader cross-category generalization is an important next step and proposed to run such experiments (for example filter to sort or GitLab to Reddit) if needed to strengthen the revision.

- ****Stronger Base Models and Performance Versus Cost Analysis (All Reviewers)**** Reviewers requested experiments with more powerful models and an assessment of inference cost. We added experiments with GPT-5 as both base agent and hinter, and with a vision-based CUA agent combined with Claude-4.5-Sonnet. **DR.Hinter continues to provide substantial improvements in all cases, confirming its value even with state-of-the-art models.** We also added a performance-versus-cost analysis. DR.Hinter’s hint retrieval and consumption introduce minimal inference overhead relative to ReAct baselines. All heavy computation occurs offline during hint generation. The revised paper shows that DR.Hinter improves accuracy with essentially no increase in decision-time cost.

- ****Hint Quality Control and Abstraction (bPLF, iAEN)**** Reviewers requested clarification on handling duplicates, contradictions, or overly task-specific hints. We added a Curator module that merges semantically similar hints, removes duplicates, and produces generalized instructions. While performance gains are small due to the base agent’s intrinsic filtering, **this improves consistency and reduces noise.** The revised manuscript documents the Curator’s behavior.


[1] Holt, S., Luyten, M. R., & Pouplin, T. (2025). Improving LLM Agent Planning with In-Context Learning via Atomic Fact Augmentation and Lookahead Search. ArXiv.
[2] Minghao Chen, Yihang Li, et al. AutoManual: Constructing Instruction Manuals by LLM Agents via Interactive Environmental Learning.

---

Across the discussion period, we added new ablations, stronger-model experiments, a performance-versus-cost analysis, curator results, expanded comparisons to prior work, and detailed evidence for the impact of zooming. These additions directly address the main concerns raised. We hope this summary assists the AC in evaluating the revised submission.

---

### Meta-Review · Area_Chair_v2So · 2026-01-07

**Summary:**

This paper presents DR.Hinter, an offline pipeline that distills successful and failed web-agent trajectories into retrievable natural-language hints with a zooming mechanism. Reviewers agree the system is clear, interpretable, and empirically strong on MiniWoB++, WorkArena, and WebArena-Lite, including useful ablations and stronger-model checks. However, the core ideas feel incremental relative to prior hint/manual/memory work, and much of the contribution is prompt engineering and engineering choices. Evidence for broad generalization beyond closely related browser tasks/templates remains limited, and more direct comparisons/analysis of hint quality would strengthen the case. Overall, I recommend rejection.

**Reviewer Concerns:**

- Rebuttal addressed: (1) clearer evidence that zooming matters, (2) added stronger-model results, (3) some mitigation for duplicate/contradictory hints (curation/filtering).

- Still outstanding: (1) novelty remains incremental, (2) generalization is largely within same web/task distributions (template similarity), (3) ross-domain transfer/robust reproducibility under prompt sensitivity isn’t convincingly shown.

**Reviewer Scores:**

- bPLF: likely would have raised their score a bit after the added zooming analysis, clearer LLM/non-LLM breakdown, and stronger-model results, but novelty concerns would likely still cap the score.
- YdbX: likely keep the score or raise a bit after the additional experiments/clarifications.
- iAEN: maybe keep the same rating even with fuller discussion.

---

### Decision · Program_Chairs · 2026-01-26

Reject